# A Fair Generative Model
# Using Total Variation Distance

## Abstract

We explore a fairness-related challenge that arises in generative models. The challenge is that biased training data with imbalanced representations of demographic groups may yield a high asymmetry in size of generated samples across distinct groups. We focus on practically-relevant scenarios wherein demographic labels are not available and therefore the design of a fair generative model is particularly challenging. In this paper, we propose an optimization framework that regulates the unfairness under such practical settings by employing one prominent statistical notion, total variation distance (TVD). We quantify the degree of unfairness via the TVD between the generated samples and balanced-yet-small reference samples. We take a variational optimization approach to faithfully implement the TVD-based measure. Experiments on benchmark real datasets demonstrate that the proposed framework can significantly improve the fairness performance while maintaining realistic sample quality for a wide range of the reference set size all the way down to 1% relative to training set.

## 1 Introduction

High-quality realistic samples synthesized thanks to recent advances in generative models (Brock et al., 2019; Goodfellow et al., 2014; Karras et al., 2019) have played a crucial role to enrich training data for a widening array of applications such as face recognition, natural language processing, and medical imaging (Wang et al., 2019; Chang et al., 2018; Yi et al., 2019). One challenge concerning *fairness* arises when generative models are built upon biased training data that preserve unbalanced representations of demographic groups. Any existing bias in the dataset can readily be propagated to the learned model, thus producing generations that are biased towards certain demographics. The unbalanced generated samples may often yield undesirable performances against underrepresented groups for downstream applications. One natural way to ensure fair sample generation is to exploit demographic labels (if available) to build a fair generative model, e.g., via conditional GAN (Mirza & Osindero, 2014; Odena et al., 2017; Miyato & Koyama, 2018) which employs such labels to easily generate an arbitrary number of samples for minority groups. In many practically-relevant scenarios, however, such labels are not often available.

To address the challenge, one pioneering work (Choi et al., 2020) develops a novel debiasing technique that employs the *reweighting* idea (Ren et al., 2018; Kamiran & Calders, 2012; Byrd & Lipton, 2019) to put more weights to underrepresented samples, thereby promoting fair sample generation across demographic groups. One key feature of the technique is to identify the bias (reflected in the weights) via a small and unlabelled reference dataset. While it enjoys significant fairness performance for moderate sizes of the reference dataset, it may provide a marginal gain for a more practically-relevant case of a small set size where the weight estimation is often inaccurate, as hinted by the meta-learning literature (Ren et al., 2018; Shu et al., 2020). We also find such phenomenon in our experiments; see Table 2 for details.

On the other hand, one recent study (Roh et al., 2020) sheds lights on addressing the small set size issue. Roh et al. (2020) propose a robust training approach by employing the Jensen-Shannon divergence (or equivalently mutual information (Majtey et al., 2005)) between poisoned training and clean reference samples. It then takes the divergence as a regularization term in order to promote robust training. One key benefit of Roh et al. (2020)in light of the reweighting-based approaches (Ren et al., 2018; Choi et al., 2020) is that the robustness performance is guaranteed even for a small

size of the clean reference set down to 5% relative to the training set size. This implies that the divergence-based regularization approach makes a more efficient use of reference data for robustness, as compared to the reweighting technique.

**Contribution:** Inspired by the key benefit featured in Roh et al. (2020), we address the small set issue by another well-known statistical measure, in particular, *total variation distance (TVD)*. See Section 3.2 for details on the rationale behind the use of TVD. Similarly we introduce a reference dataset which is balanced and unlabelled. We then employ it to formulate the TVD between the generated and reference sample distributions, which can serve as quantifying the degree of unfairness. We then promote fair sample generation by adding the TVD into a conventional optimization (e.g., GAN-based optimization (Goodfellow et al., 2014; Nowozin et al., 2016; Arjovsky et al., 2017)). Motivated by the variational optimization technique w.r.t. TVD (Villani, 2009; Nowozin et al., 2016; Arjovsky et al., 2017), we translate the TVD-regularization term into a function optimization. We also conduct extensive experiments on three benchmark real datasets: CelebA (Liu et al., 2015), UTKFace (Zhang et al., 2017), and FairFace (Karkkainen & Joo, 2021). We demonstrate via simulation that the proposed framework can significantly boost up the fairness performance while offering high-quality realistic samples reflected in low FID. We also find that our approach outperforms the state of the art (Choi et al., 2020), particularly being robust to the balanced reference set size: the significant improvements preserve for a wide range of the reference set size down to 1% relative to training data (more preferable in reality).

**Related works:** In addition to Choi et al. (2020), Tan et al. (2020) propose a different way that promotes fair sample generation by smartly perturbing the input distribution of a pre-trained generative model with the help of a classifier for sensitive attributes. The key distinction w.r.t. ours is that it relies upon the additional classifier which requires the use of demographic labels to obtain. Yu et al. (2020) employ demographic labels for minority groups to generate a wide variety of samples with improved data converge by harmonizing GAN and MLE ideas. A distinction w.r.t. ours is that it requires the knowledge on demographic labels. Jalal et al. (2021) consider a fair generative model yet in a different context, image reconstruction. The goal of the task is to ensure fair sample generation of restored images from degraded versions. Since it relies upon the degraded images, it is not directly comparable to ours. Another line of fair generative modeling focuses on *label* bias, instead of representation bias (Xu et al., 2018; 2019a;b; Sattigeri et al., 2019; Jang et al., 2021; Kyono et al., 2021). The goal therein is to develop a generative model such that the generated decision labels are statistically independent of the given demographic labels. Again, these are not directly comparable to ours, as they require the use of demographic labels.

The variational optimization technique w.r.t. TVD that gives an inspiration to our work has originated from Villani (2003; 2009), wherein the author shows that the TVD can be expressed as a function optimization in the context of transport theory. The technique was recently applied to the GAN context (Nowozin et al., 2016). The TVD has also served as a useful tool for quantifying various fairness measures in fair classifiers that pursue individual fairness (Dwork et al., 2012; Dwork & Ilvento, 2018) and group fairness (Gordaliza et al., 2019; Wang et al., 2020a; Farokhi, 2021).

## 2 PROBLEM FORMULATION

**Setup:** Fig. 1 illustrates the problem setting for a fair generative model that we focus on herein. We consider a challenging yet practically-relevant scenario wherein demographic information (or that we call sensitive attribute), say $z \in \mathcal{Z}$, is not available. Under this blind setting, the goal of the problem is to construct a *fair* generative model that ensures the produced samples to have the same size across distinct demographics. We assume that there are two types of data given in the problem: (i) training data $\mathcal{D}_{\text{bias}} := \{x_{\text{bias}}^{(i)}\}_{i=1}^{m_{\text{bias}}}$; (ii) reference data $\mathcal{D}_{\text{ref}} := \{x_{\text{ref}}^{(i)}\}_{i=1}^{m_{\text{ref}}}$. Since we consider the setting where training data is potentially biased, we use the word "bias" in the associated notations. Here $m_{\text{bias}}$ denotes the number of training examples. Let $\mathbb{P}_{\text{bias}}$ be data distribution which each training data $x_{\text{bias}}^{(i)} \in \mathcal{X}$ is generated upon. In a biased scenario having female-vs-male sensitive attribute, e.g., $z = 0$ (female) and $z = 1$ (male), we may have $\mathbb{P}_{\text{bias}}(Z = 0) > \mathbb{P}_{\text{bias}}(Z = 1)$. As in Choi et al. (2020), we also employ a balanced yet small reference dataset for the purpose of promoting fair sample generation, which can be obtained via a set of carefully-designed data collection protocols employed in organizations like World Bank and biotech companies (Choi et al., 2020; 23&me, 2016; Hong, 2016). Let $\mathbb{P}_{\text{ref}}$ be the corresponding data distribution defined on $\mathcal{X}$:

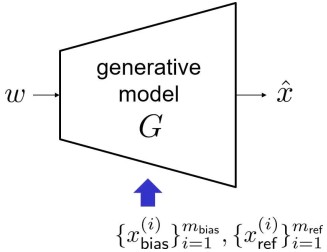

Figure 1: Part of a fair generative model that intends to yield generated samples with the equal size over demographic groups. We employ training data $\{x_{\mathsf{bias}}^{(i)}\}_{i=1}^{m_{\mathsf{bias}}}$ (potentially biased) and balanced reference data $\{x_{\mathsf{ref}}^{(i)}\}_{i=1}^{m_{\mathsf{ref}}}$. The entire structure of the proposed model will be illustrated in detail in Fig. 2. Here $m_{\mathsf{bias}}$ (or $m_{\mathsf{ref}}$) denotes the number of training samples (or reference samples).

$\mathbb{P}_{\mathsf{ref}}(Z = 0) \approx \mathbb{P}_{\mathsf{ref}}(Z = 1)$. Since the balanced reference set is often challenging to collect, typically the number of the reference samples is much smaller than that of training examples: $m_{\mathsf{ref}} \ll m_{\mathsf{bias}}$. Denote by $\hat{x} := G(w) \in \mathcal{X}$ the generated sample fed by a random noise input $w \in \mathcal{W}$. We assume that the generated samples have the same support $\mathcal{X}$ as training and reference samples. Let $\mathbb{P}_G$ and $\mathbb{P}_W$ be distributions w.r.t. generated samples and the random noise input respectively.

As a fairness measure that will be employed for the purpose of evaluating our framework to be presented in Section 3, we consider *fairness discrepancy* proposed by Choi et al. (2020). It quantifies how $\mathbb{P}_G$ differs from $\mathbb{P}_{\mathsf{ref}}$ w.r.t. a certain sensitive attribute, formally defined below.

**Definition 1** (Fairness Discrepancy (Choi et al., 2020))**.** *Fairness discrepancy between $\mathbb{P}_{\mathsf{ref}}$ and $\mathbb{P}_G$ w.r.t. a sensitive attribute $z \in \{z_1, \ldots, z_{|\mathcal{Z}|}\}$ is defined as:*

$$\mathcal{F}(\mathbb{P}_{\mathsf{ref}}, \mathbb{P}_G) := \|\mathbf{p}_{\mathsf{ref}}(z) - \mathbf{p}_G(z)\|_2 \tag{1}$$

*where*

$$\mathbf{p}_{\mathsf{ref}}(z) := \begin{bmatrix} \mathbb{P}_{\mathsf{ref}}(Z = z_1) \\ \mathbb{P}_{\mathsf{ref}}(Z = z_2) \\ \vdots \\ \mathbb{P}_{\mathsf{ref}}(Z = z_{|\mathcal{Z}|}) \end{bmatrix} \quad and \quad \mathbf{p}_G(z) := \begin{bmatrix} \mathbb{P}_G(\hat{Z} = z_1) \\ \mathbb{P}_G(\hat{Z} = z_2) \\ \vdots \\ \mathbb{P}_G(\hat{Z} = z_{|\mathcal{Z}|}) \end{bmatrix}.$$

Here $\hat{Z}$ denotes the prediction of the sensitive attribute w.r.t. a generated sample, yielded by a pre-trained classification model which we call *attribute classifier* as in Choi et al. (2020). The attribute classifier is employed only for the purpose of evaluation, and is trained based on another real dataset, e.g., like the one mentioned in Choi et al. (2020): the standard train and validation splits of CelebA (Liu et al., 2015). For faithful evaluation, we consider a vast number of generated samples (i.e., more than 10,000) as well as employ highly-accurate attribute classifiers, around 98% accuracy of gender classifier for instance.

As a measure for the quality of generated samples that may compete with the fairness measure, we employ a well-known measure: Fréchet Inception Distance (FID) (Heusel et al., 2017). It is defined as the Fréchet distance (Fréchet, 1957) (also known as the second-order Wasserstein distance (Wasserstein, 1969)) between real and generated samples approximated via the Gaussian distribution. The lower FID, the more realistic and diverse the generated samples are. For a more precise measure that reveals sample quality of each sensitive group, we consider FID computed *within* each demographic, called *intra* FID (Miyato & Koyama, 2018; Zhang et al., 2019; Wang et al., 2020b). Computing intra FID requires the knowledge on group identities of generated samples. Since demographic labels are not available in our context, we rely upon the attribute classifier (that we introduced above) for predicting demographic information of the generated samples.

**GAN-based generative model:** Our framework (to be presented soon) builds upon one powerful generative model: Generative Adversarial Network (GAN) (Goodfellow et al., 2014). The GAN comprises two competing players: (i) discriminator $D(\cdot)$ that wishes to discriminate real samples against generated samples; and (ii) generator $G(\cdot)$ that intends to fool the discriminator by producing

realistic generated samples. In particular, we consider a general optimization framework (Tseng et al., 2021) which subsumes many GAN variants as special cases:

$$\begin{aligned} \textit{(Discriminator)} \quad & \max_D \ \mathbb{E}_{\mathbb{P}_{\text{bias}}}\big[f_D(D(X))\big] + \mathbb{E}_{\mathbb{P}_G}\big[f_G(D(X))\big]; \\ \textit{(Generator)} \quad & \min_G \ -\mathbb{E}_{\mathbb{P}_G}\big[g_G(D(X))\big] \end{aligned} \tag{2}$$

where $f_D$, $f_G$, and $g_G$ indicate certain functions that vary depending on an employed GAN approach. For instance, the choice of $(f_D(t), f_G(t), g_G(t)) = (t, -t, t)$ together with Lipschitz-1 condition on $D$ leads to the prominent WGAN optimization (Arjovsky et al., 2017). Another choice of $(f_D(t), f_G(t), g_G(t)) = (\min\{0, -1 + t\}, \min\{0, -1 - t\}, t)$ yields a hinge-loss-based GAN (Lim & Ye, 2017; Tran et al., 2017). We adopt this as a base framework that we will add a fairness aspect into in the next section.

## 3 PROPOSED FRAMEWORK

### 3.1 TVD-BASED APPROACH

One natural way to enforce a fairness constraint is to incorporate a fairness-regularization term into the base framework. However, since the base framework consists of *two distinct* optimizations, how to add the regularization term is not clear. To gain some insights into this, we focus on one instance of GAN, which allows us to express the two as only one optimization and then relate the optimization to a well-known statistical notion: total variation distance (TVD). To this end, we first consider the following mappings: $(f_D(t), f_G(t), g_G(t)) = (t, -t, t)$. Applying these into equation 2 gives:

$$\min_G \max_D \ \mathbb{E}_{\mathbb{P}_{\text{bias}}}\big[D(X)\big] - \mathbb{E}_{\mathbb{P}_G}\big[D(X)\big]. \tag{3}$$

We now employ the variational optimization technique presented in Villani (2003; 2009); Nowozin et al. (2016); Arjovsky et al. (2017) to translate equation 3 into the TVD between $\mathbb{P}_{\text{bias}}$ and $\mathbb{P}_G$. Details are described in Theorem 1 below.

**Theorem 1** (Nowozin et al. (2016)). *The optimization in equation 3 with the bounded constraint* $D(\cdot) \in [-1, 1]$ *is equivalent to:*

$$\min_G \ \mathsf{TV}(\mathbb{P}_{\text{bias}}, \mathbb{P}_G) \tag{4}$$

*where*

$$\mathsf{TV}(\mathbb{P}_{\text{bias}}, \mathbb{P}_G) := \frac{1}{2} \sum_{x \in \mathcal{X}} |\mathbb{P}_{\text{bias}}(x) - \mathbb{P}_G(x)|.$$

*Proof.* See appendix A. □

### 3.2 FAIR GENERATIVE MODEL

Now we have the TVD-based framework with the *single* optimization. So one natural way to impose a fairness constraint is to add a proper regularization term on top of equation 4. Remember our framework employs the reference dataset containing balanced samples, reflected in $\mathbb{P}_{\text{ref}}$. Hence, as the regularization term, we may consider a distance measure between $\mathbb{P}_{\text{ref}}$ (target distribution) and $\mathbb{P}_G$, since a large distance penalizes the objective. To this end, we propose to use the same TVD as was used in equation 4:

$$\min_G \ (1 - \lambda) \cdot \mathsf{TV}(\mathbb{P}_{\text{bias}}, \mathbb{P}_G) + \lambda \cdot \mathsf{TV}(\mathbb{P}_{\text{ref}}, \mathbb{P}_G) \tag{5}$$

where $\lambda \in [0, 1]$ denotes a normalized regularization factor that balances the sample quality against the fairness constraint. The rationale behind the use of TVD regularization is that it is robust to the size of dataset compared to a set of $f$-divergences (Tseng et al., 2021). Moreover, it offers the best trade-off performances in fairness and sample quality compared to other divergence measures: Jensen-Shannon divergence (Wong & You, 1985), Kullback-Leibler divergence (Kullback & Leibler, 1951), Pearson $\chi^2$-divergence (Pearson, 1900), and Wasserstein distance (Wasserstein, 1969). We

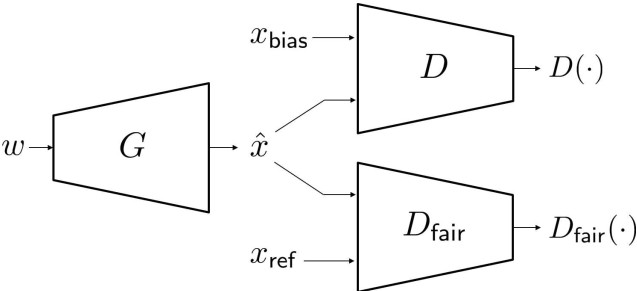

Figure 2: The architecture of the proposed three-player optimization, reflected in equation 6.

provide empirical evidence that demonstrates this performance benefit in Section 4.2; see Table 3 for details.

One challenge that arises in equation 5 is that expressing the TVDs in terms of an optimization variable $G$ is not that straightforward. For ease of implementation, we rely upon Theorem 1 again, which shows the equivalence between the TVD-based optimization and the two-player game (equation 3). Since equation 5 includes one more TVD, we introduce the third player, who serves as another discriminator trying to distinguish generated samples from the balanced reference samples. Based on this, we propose a three-player game which is equivalent to equation 5. See Theorem 2 for the formal statement of the equivalence and its proof.

**Theorem 2.** *The optimization in equation 5 is equivalent to:*

$$\min_{G} \max_{|D_{\mathsf{fair}}| \leq 1} \max_{|D| \leq 1} (1-\lambda)\{\mathbb{E}_{\mathbb{P}_{\mathsf{bias}}}[D(X)] - \mathbb{E}_{\mathbb{P}_G}[D(X)]\} + \lambda\{\mathbb{E}_{\mathbb{P}_{\mathsf{ref}}}[D_{\mathsf{fair}}(X)] - \mathbb{E}_{\mathbb{P}_G}[D_{\mathsf{fair}}(X)]\}.$$
(6)

*Proof.* Manipulating equation 6, we obtain:

$$\min_{G} \max_{|D_{\mathsf{fair}}| \leq 1} \max_{|D| \leq 1} (1-\lambda)\{\mathbb{E}_{\mathbb{P}_{\mathsf{bias}}}[D(X)] - \mathbb{E}_{\mathbb{P}_G}[D(X)]\} + \lambda\{\mathbb{E}_{\mathbb{P}_{\mathsf{ref}}}[D_{\mathsf{fair}}(X)] - \mathbb{E}_{\mathbb{P}_G}[D_{\mathsf{fair}}(X)]\}$$
$$= \min_{G} (1-\lambda) \sum_{x \in \mathcal{X}} \{\mathbb{P}_{\mathsf{bias}}(x) - \mathbb{P}_G(x)\} D^*(x) + \lambda \sum_{x \in \mathcal{X}} \{\mathbb{P}_{\mathsf{ref}}(x) - \mathbb{P}_G(x)\} D^*_{\mathsf{fair}}(x)$$
(7)

where the equality is because we assume the same support $\mathcal{X}$ for all data samples. Under bounded discriminator functions $D(\cdot), D_{\mathsf{fair}}(\cdot) \in [-1, 1]$, one can readily obtain the optimal discriminators as:

$$D^*(x) = \mathsf{sign}\{\mathbb{P}_{\mathsf{bias}}(x) - \mathbb{P}_G(x)\};$$
$$D^*_{\mathsf{fair}}(x) = \mathsf{sign}\{\mathbb{P}_{\mathsf{ref}}(x) - \mathbb{P}_G(x)\}.$$

Substituting these into equation 7, we get equation 5. This completes the proof. □

The optimization structure of the new discriminator $D_{\mathsf{fair}}$ is the same as that of $D$, except that we read $\mathbb{P}_{\mathsf{ref}}$ instead of $\mathbb{P}_{\mathsf{bias}}$. The generator optimization has an additional term w.r.t. $D_{\mathsf{fair}}(X)$ accordingly. The translated three-player game can then be implemented, e.g., via parameterization of three neural networks. We then employ a three-way alternating gradient descent (Goodfellow et al., 2014) for the parameterized neural networks. This procedure is formally presented in Algorithm 1.

**Remark 1** (Implementing bounded functions via hinge loss). *Among a multitude of practices for implementing bounded functions $D(\cdot), D_{\mathsf{fair}}(\cdot) \in [-1, 1]$, we take an approximated approach that offers empirical benefits: the use of hinge loss (Lim & Ye, 2017; Tran et al., 2017). As expressed in Steps 7 and 12 in Algorithm 1, we apply the hinge loss in both $D$ and $D_{\mathsf{fair}}$ so that the discriminators attempt to constrain themselves within the interval $[-1, 1]$. We found via extensive experiments that this choice enables more stabilized training, thereby offering greater performances relative to a direct and simple approach: employing tanh activations in $D$ and $D_{\mathsf{fair}}$. Actually this hinge-loss-based approximation is inspired by Nguyen et al. (2009); Arjovsky et al. (2017); Tan et al. (2019) which explore in depth the equivalence between TVD and hinge-loss-based optimization.* ∎

---

**Algorithm 1** Three-player optimization for the TVD-based fair generative model

---

1: **Input:** Training data $\mathcal{D}_{\mathsf{bias}} = \{x_{\mathsf{bias}}^{(i)}\}_{i=1}^{m_{\mathsf{bias}}}$, reference data $\mathcal{D}_{\mathsf{ref}} = \{x_{\mathsf{ref}}^{(i)}\}_{i=1}^{m_{\mathsf{ref}}}$, noise input distribution $\mathbb{P}_W$, and several hyperparameters: $\lambda$; the numbers of discriminator steps $k$ and $l$; and the number of iterations $T$

2: **Initialize:** $G(w; \theta)$ (generator), $D(x; \phi)$ (discriminator), and $D_{\mathsf{fair}}(x; \psi)$ (fairness discriminator)

3: **for** $T$ steps **do**

4:     **for** $k$ steps **do** {*Loop for training $D(x; \phi)$*}

5:         Sample minibatch of $m_{\mathcal{B}}$ noise inputs $\{w^{(1)}, \ldots, w^{(m_{\mathcal{B}})}\}$ from $\mathbb{P}_W$

6:         Sample minibatch of $m_{\mathcal{B}}$ training data points $\{x_{\mathsf{bias}}^{(1)}, \ldots, x_{\mathsf{bias}}^{(m_{\mathcal{B}})}\}$ from $\mathcal{D}_{\mathsf{bias}}$ at random

7:         Update $D(x; \phi)$ by ascending its stochastic gradient:

$$\nabla_\phi \frac{1-\lambda}{m_{\mathcal{B}}} \sum_{i=1}^{m_{\mathcal{B}}} \left[ \min\left\{0, -1 + D(x_{\mathsf{bias}}^{(i)}; \phi)\right\} + \min\left\{0, -1 - D(G(w^{(i)}; \theta); \phi)\right\} \right]$$

8:     **end for**

9:     **for** $l$ steps **do** {*Loop for training $D_{\mathsf{fair}}(x; \psi)$*}

10:         Sample minibatch of $m_{\mathcal{B}_{\mathsf{fair}}}$ noise inputs $\{w^{(1)}, \ldots, w^{(m_{\mathcal{B}_{\mathsf{fair}}})}\}$ from $\mathbb{P}_W$

11:         Sample minibatch of $m_{\mathcal{B}_{\mathsf{fair}}}$ reference data points $\{x_{\mathsf{ref}}^{(1)}, \ldots, x_{\mathsf{ref}}^{(m_{\mathcal{B}_{\mathsf{fair}}})}\}$ from $\mathcal{D}_{\mathsf{ref}}$ at random

12:         Update $D_{\mathsf{fair}}(x; \psi)$ by ascending its stochastic gradient:

$$\nabla_\psi \frac{\lambda}{m_{\mathcal{B}_{\mathsf{fair}}}} \sum_{i=1}^{m_{\mathcal{B}_{\mathsf{fair}}}} \left[ \min\left\{0, -1 + D_{\mathsf{fair}}(x_{\mathsf{ref}}^{(i)}; \psi)\right\} + \min\left\{0, -1 - D_{\mathsf{fair}}(G(w^{(i)}; \theta); \psi)\right\} \right]$$

13:     **end for**

14:     Sample minibatch of $m_{\mathcal{B}_G}$ noise inputs $\{w^{(1)}, \ldots, w^{(m_{\mathcal{B}_G})}\}$ from $\mathbb{P}_W$

15:     Update $G(w; \theta)$ by descending its stochastic gradient:

$$\nabla_\theta \left[ -\frac{1-\lambda}{m_{\mathcal{B}_G}} \sum_{i=1}^{m_{\mathcal{B}_G}} D(G(w^{(i)}; \theta); \phi) - \frac{\lambda}{m_{\mathcal{B}_G}} \sum_{j=1}^{m_{\mathcal{B}_G}} D_{\mathsf{fair}}(G(w^{(j)}; \theta); \psi) \right]$$

16: **end for**

---

**Remark 2** (Three-way battles). *Fig. 2 illustrates the entire architecture of the translated three-level optimization. Here we see interesting three-way battles. The first is a well-known battle between the generator $G$ and the 1st discriminator $D$. Remember $D^*(x) = \mathsf{sign}\{\mathbb{P}_{\mathsf{bias}}(x) - \mathbb{P}_G(x)\}$. So one can interpret $D^*$ as the strength of distinguishing real (potentially biased) samples against generated samples. On the other hand, the generator intends to fool $D$, thus promoting realistic samples. The second battle is in between the generator and the 2nd discriminator $D_{\mathsf{fair}}$. The same interpretation can be made from $D^*_{\mathsf{fair}}(x) = \mathsf{sign}\{\mathbb{P}_{\mathsf{ref}}(x) - \mathbb{P}_G(x)\}$ (the ability to distinguish balanced reference samples against the generated samples). This way, the generator $G$ is encouraged to produce* balanced *yet less realistic (due to the small-sized reference set) samples, thus pitting the 1st discriminator against the 2nd discriminator indirectly. The last battle is in between the 1st and 2nd discriminators. This tension is* directly *controlled by the fairness tuning knob $\lambda$; see corresponding tradeoff curves presented in Fig. 4 in appendix E.4. It turns out the three-way tradeoff relationships established via our TVD-based framework are greatly balanced, thus achieving significant performances both in fairness and sample quality. This is empirically demonstrated in Section 4.2; see Tables 1 and 3 for details.* ∎

## 4 EXPERIMENTS

We conduct experiments on three benchmark real datasets: CelebA (Liu et al., 2015), UTK-Face (Zhang et al., 2017), and FairFace (Karkkainen & Joo, 2021). We implement our algorithm in PyTorch (Paszke et al., 2019), and all experiments are performed on servers with TITAN RTX and

Quadro RTX 8000 GPUs. For our algorithm, all the simulation results (to be reported) are the ones averaged over five trials with distinct seeds.

## 4.1 SETUP

*Datasets:* Our construction of $\mathcal{D}_{\text{bias}}$ and $\mathcal{D}_{\text{ref}}$ respects the method described in Choi et al. (2020). Only for the purpose of data construction, we have an access to sensitive attributes $z$, so as to control the ratio of demographic group sizes. For CelebA, we consider two scenarios depending on the number of focused attributes: (i) CelebA-single (gender); (ii) CelebA-multi (two attributes: gender and hair color). Training data $\mathcal{D}_{\text{bias}}$ is constructed to have $9 : 1$ ratio (female vs. male) samples where $m_{\text{bias}} = 67507$. We take balanced samples for $\mathcal{D}_{\text{ref}}$ ($1 : 1$ ratio). For CelebA-multi, we have four groups: (i) (female, non-black); (ii) (male, non-black); (iii) (female, black); (iv) (male, black). For $\mathcal{D}_{\text{bias}}$, we take $85 : 15$ ratio samples (non-black hair vs. black hair) where $m_{\text{bias}} = 60000$. For UTKFace dataset, we consider a race attribute: white vs. non-white. We take $9 : 1$ ratio biased samples with $m_{\text{bias}} \approx 10000$. For FairFace dataset, we consider another type of race categorized as white vs. black. We also take the $9 : 1$ ratio biased samples yet with $m_{\text{bias}} \approx 20000$. A wide range of the reference set size is taken into consideration. We focus mainly on two sizes: (i) 10% ($m_{\text{ref}} \approx 0.1 m_{\text{bias}}$); (ii) 25% ($m_{\text{ref}} \approx 0.25 m_{\text{bias}}$). To demonstrate the robustness of our proposed approach to the reference set size, we also consider small sizes of the reference set down to 1%. See appendix B.1 for more details.

*Baselines:* We consider three baselines. The first baseline, say Baseline I, is a *non-fair* algorithm building upon the base framework described in equation 2. It is trained on the aggregated dataset $\mathcal{D}_{\text{bias}} \cup \mathcal{D}_{\text{ref}}$. The second baseline, say Baseline II, is the same non-fair algorithm yet trained only with a small balanced reference set $\mathcal{D}_{\text{ref}}$. The last is the state of the art, Choi et al. (2020). For all three baselines, we employ the hinge loss optimization (Lim & Ye, 2017; Tran et al., 2017): choosing the mapping functions in equation 2 as $(f_D(t), f_G(t), g_G(t)) = (\min\{0, -1 + t\}, \min\{0, -1 - t\}, t)$.

*Attribute classifiers:* As mentioned in Section 2 (near Definition 1), we employ attribute classifiers, only for the purpose of evaluating our twin measures: (i) fairness discrepancy (defined in equation 1); (ii) intra FID. We introduce four different attribute classifiers for predicting senstive attributes in the following scenarios: (i) gender for CelebA-single; (ii) gender and hair-color for CelebA-multi; (iii) white-vs-non-white race for UTKFace; (iv) white-vs-black race for FairFace. For all the classifiers, we use a variant of ResNet18 (He et al., 2016). CelebA and FairFace classifiers are trained over the standard train and validation splits of CelebA (Liu et al., 2015) and FairFace (Karkkainen & Joo, 2021), respectively. For training UTKFace classifier, we use $8 : 1 : 1$ splits of UTKFace (Zhang et al., 2017) dataset. We found that our evaluation is sensitive to the performances of the attribute classifiers; see appendix C.1 for a detailed discussion.

*Hyperparameter search:* For implementation of all three baselines (Baselines I, Baseline II, and Choi et al. (2020)) and the proposed framework, we all employ the BigGAN architecture (Brock et al., 2019). In other words, we parameterize $G$, $D$, and $D_{\text{fair}}$ with the neural-net architecture introduced in Brock et al. (2019). We leave details in appendix B.2. We also provide a complexity analysis of our algorithm with a comparison to the state of the art (Choi et al., 2020); see appendix D for details.

## 4.2 RESULTS

Table 1 provides performance comparison with the three baselines on CelebA dataset. Notice for all the considered settings that our approach exhibits better (or comparable) performances than the state of the art (Choi et al., 2020) both in fairness ("Fairness discrepancy") and sample quality ("Intra FID"). The lower, the better for all the measures. The performance gaps are more apparent in the small reference set size. Notice in the last row w.r.t. CelebA-single in Table 1 that the fairness performance exhibits slight degradation with an increase in the reference set size. See appendix C.2 for an in-depth discussion on this counter-intuitive adverse effect of the increased reference set size. Due to space limitation, we leave in appendix E.1 experimental results conducted on the other two datasets: UTKFace and FairFace. The performance trends are similar to those in CelebA.

Fig. 3 visualizes generated samples on CelebA-single with the 10% reference set size. The top figure corresponds to Choi et al. (2020), while the bottom is due to the proposed algorithm. For each figure, faces above the yellow lines are female samples, while the rest are male samples. Here the gender is

Table 1: Performance comparison on CelebA dataset. CelebA-single indicates the scenario in which a *single* attribute (gender) is employed. CelebA-multi refers to the case considering two attributes: gender and hair color. Baseline I is a non-fair algorithm building upon the base framework with the hinge loss (Lim & Ye, 2017; Tran et al., 2017), and trained with the aggregated data $\mathcal{D}_{\text{bias}} \cup \mathcal{D}_{\text{ref}}$. Baseline II is the same non-fair algorithm yet trained only with the small yet balanced reference dataset $\mathcal{D}_{\text{ref}}$. Choi et al. is the state of the art (Choi et al., 2020). "Intra FID" refers to Fréchet Inception Distance (Heusel et al., 2017) computed within each group (Miyato & Koyama, 2018; Zhang et al., 2019), and we provide results for minority groups herein; see appendix E.2 for complete results. The lower intra FID, the more realistic and diverse samples. "Fairness" is *fairness discrepancy* introduced by Choi et al. (2020); see equation 1 for the definition. The lower, the fairer. The reference set size indicates a ratio relative to training data.

| | | CelebA-single | | CelebA-multi | |
|---|---|---|---|---|---|
| Reference set size | | 10% | 25% | 10% | 25% |
| Baseline I | Intra FID | $12.73 \pm 0.053$ | $12.00 \pm 0.069$ | $10.83 \pm 0.046$ | $9.76 \pm 0.067$ |
| | Fairness | $0.554 \pm 0.002$ | $0.495 \pm 0.001$ | $0.245 \pm 0.002$ | $0.224 \pm 0.002$ |
| Baseline II | Intra FID | $32.31 \pm 0.109$ | $23.81 \pm 0.118$ | $29.66 \pm 0.088$ | $21.92 \pm 0.056$ |
| | Fairness | $0.115 \pm 0.002$ | $0.093 \pm 0.002$ | $0.136 \pm 0.001$ | $0.107 \pm 0.001$ |
| Choi et al. | Intra FID | $25.74 \pm 0.079$ | $20.68 \pm 0.076$ | $16.27 \pm 0.042$ | $13.68 \pm 0.062$ |
| | Fairness | $0.104 \pm 0.002$ | $0.065 \pm 0.002$ | $0.090 \pm 0.001$ | $0.063 \pm 0.002$ |
| Proposed | Intra FID | $14.29 \pm 1.354$ | $11.34 \pm 1.288$ | $12.93 \pm 0.631$ | $12.07 \pm 0.299$ |
| | Fairness | $0.057 \pm 0.012$ | $0.069 \pm 0.015$ | $0.080 \pm 0.006$ | $0.073 \pm 0.002$ |

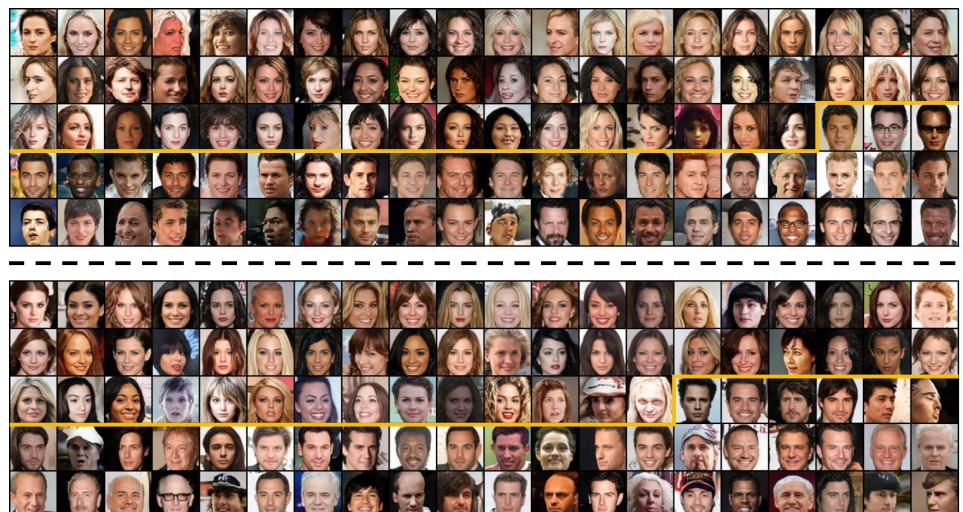

Figure 3: (Top) Generated samples by Choi et al. (2020) trained on CelebA-single with 10% reference set size. Faces above the yellow line are female (57 pictures), while the rest are male (43). Intra FID values are around 21.07 (female) and 25.74 (male); (Bottom) Generated samples by the proposed approach under the same setting. We obtain 54 females and 46 males, yet producing more realistic sample images, reflected in a much lower intra FID values, around 9.51 (female) and 14.29 (male).

predicted via the attribute classifier. While both approaches yield well-balanced samples ($57 : 43$ for Choi et al. (2020), and $54 : 46$ for ours; see Table 1 for a more precise performance comparison), our algorithm produces more realistic sample images. This is reflected in lower intra FID values, around $9.51$ (female group) and $14.29$ (male group). On the other hand, Choi et al. (2020) offers intra FID values of around $21.07$ and $25.74$ for female and male groups, respectively. See Table 17 (in appendix E) for the values of intra FID. In appendix F, we also provide generated samples for UTKFace and FairFace datasets.

Table 2: Robustness of the proposed approach to the reference set size down to $1\%$. This is evaluated on CelebA-single. All the other settings and baselines are the same as those in Table 1. Our algorithm offers more robust performances both in fairness and intra FID.

| Reference set size | | 5% | 2.5% | 1% |
|---|---|---|---|---|
| Baseline I | Intra FID | $13.54 \pm 0.074$ | $13.79 \pm 0.072$ | $15.89 \pm 0.094$ |
| | Fairness | $0.559 \pm 0.001$ | $0.566 \pm 0.002$ | $0.576 \pm 0.002$ |
| Baseline II | Intra FID | $40.07 \pm 0.062$ | $67.70 \pm 0.112$ | $92.34 \pm 0.131$ |
| | Fairness | $0.120 \pm 0.003$ | $0.150 \pm 0.003$ | $0.455 \pm 0.002$ |
| Choi et al. | Intra FID | $32.30 \pm 0.049$ | $25.81 \pm 0.043$ | $30.28 \pm 0.085$ |
| | Fairness | $0.043 \pm 0.002$ | $0.274 \pm 0.002$ | $0.355 \pm 0.002$ |
| Proposed | Intra FID | $23.42 \pm 1.312$ | $24.53 \pm 1.157$ | $30.93 \pm 2.196$ |
| | Fairness | $0.048 \pm 0.012$ | $0.119 \pm 0.011$ | $0.166 \pm 0.011$ |

Table 2 demonstrates the robustness of our algorithm to the reference set size. Notice even for the 1% reference set size, our algorithm offers still respectable performances both in fairness and sample quality. On the other hand, Choi et al. (2020) suffers from performance degradation starting from 2.5%. Refer to appendix E.2 for intra FIDs w.r.t. the female group.

Table 3 demonstrates the rationale behind the use of TVD-based regularization. Observe that among the considered regularization methods, our TVD-based approach offers the best (or the second best) performances both in fairness and sample quality. It also yields the smallest discrepancy between intra FIDs of different groups. Another noticeable observation is that our divergence-based regularization approach outperforms Choi et al. (2020) for a variety of other divergence measures not limited to TVD; also see Table 1 for detailed comparison. For implementing other measures, e.g., JSD, we replace the TVD regularization term in equation 5 with the considered measure (e.g., JSD) while maintaining the first term in the objective function. We also conduct more extensive experiments which take different divergence measures even for the first term in the objective function in equation 5. Even in such cases, we obtain similar results which exhibit good performances when employing TVD for fairness regularization. See appendix E.3 for the experimental results.

Table 3: Performance comparison with other fairness regularizers on CelebA-single with the $10\%$ reference set size. "JSD-based" refers to a regularization method based on Jensen-Shannon divergence implemented via Goodfellow et al. (2014). "KLD-based" is the one built upon Kullback-Leibler divergence (Nowozin et al., 2016). "$\chi^2$-based" represents the one implementing Pearson-$\chi^2$ divergence (Mao et al., 2017). "WD-based" refers to a regularization with Wasserstein distance (Gulrajani et al., 2017). "Female" (or "Male") refers to intra FID for female (or male) group. For each measure, we mark the best result in bold and the second-best with underline.

| | JSD-based | KLD-based | $\chi^2$-based | WD-based | TVD-based |
|---|---|---|---|---|---|
| Female | $\underline{12.80 \pm 1.499}$ | $15.24 \pm 0.371$ | $16.01 \pm 1.601$ | $16.51 \pm 1.244$ | $\mathbf{9.51 \pm 1.069}$ |
| Male | $\underline{17.83 \pm 1.173}$ | $22.47 \pm 0.331$ | $25.25 \pm 1.877$ | $24.54 \pm 2.731$ | $\mathbf{14.29 \pm 1.354}$ |
| Fairness | $0.087 \pm 0.012$ | $0.077 \pm 0.019$ | $0.058 \pm 0.019$ | $\mathbf{0.047 \pm 0.033}$ | $\underline{0.057 \pm 0.012}$ |

## 5 CONCLUSION

We introduced a TVD-based optimization framework for a fair generative model that well tradeoffs the fairness performance (quantified as fairness discrepancy) against sample quality (reflected in intra FID). Inspired by the equivalence between the TVD and function optimization, we also developed an equivalent three-player optimization which can readily be implemented via neural-net parameterization. Our algorithm offers better performances than the state of the art both in fairness and intra FID, exhibiting more significant performances particularly for practically-relevant scenarios where the access to balanced dataset is limited. One future work of interest is to push forward for more challenging scenarios where the reference dataset is not available.

## ETHICS STATEMENT

One defining feature of our proposed framework is to offer fair sample generation of demographic groups while maintaining realistic high quality of generated samples. Hence, it is expected to play a crucial role to enrich well-balanced training data for a widening array of downstream applications, thereby ensuring great performances even for underrepresented groups which often suffer from degraded performances due to biased data.

One disadvantage of our approach requires the use of the *balanced reference* set. So it may not be directly applicable to more challenging scenarios where gathering such non-biased samples are very limited. Another flip side is that our framework builds upon the adversarial training approach which often incurs training instability. Hence, we believe one promising future direction is to push the boundary towards such challenging settings while addressing the training stability issue.

## REPRODUCIBILITY STATEMENT

To ensure the reproducibility of experiments, we provide a thorough description regarding our three benchmark real datasets: CelebA (Liu et al., 2015), UTKFace (Zhang et al., 2017), and Fair-Face (Karkkainen & Joo, 2021), along with data construction methods in Section 4.1 and appendix B.1. All the hyperparameter settings w.r.t. the three real datasets are presented in Appendix B.2. We also provide the average running time of our algorithm together with specific computer configuration details in Appendix D. Lastly, we provide the codes in the supplementary material.

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

## A   APPENDIX: PROOF OF THEOREM 1

**Theorem 1** (Nowozin et al. (2016)). *The optimization in equation 3 with the bounded constraint $D(\cdot) \in [-1, 1]$ is equivalent to:*

$$\min_G \ \mathsf{TV}(\mathbb{P}_{\mathsf{bias}}, \mathbb{P}_G) \tag{4}$$

*where*

$$\mathsf{TV}(\mathbb{P}_{\mathsf{bias}}, \mathbb{P}_G) := \frac{1}{2} \sum_{x \in \mathcal{X}} |\mathbb{P}_{\mathsf{bias}}(x) - \mathbb{P}_G(x)|.$$

*Proof.* We first manipulate equation 3 as:

$$
\begin{aligned}
&\min_G \max_D \ \mathbb{E}_{\mathbb{P}_{\mathsf{bias}}}\big[D(X)\big] - \mathbb{E}_{\mathbb{P}_G}\big[D(X)\big] \\
&= \min_G \ \sum_{x \in \mathcal{X}} \big\{\mathbb{P}_{\mathsf{bias}}(x) - \mathbb{P}_G(x)\big\} D^*(x)
\end{aligned} \tag{8}
$$

where the equality is because we assume the same support $\mathcal{X}$ for training and generated samples. Under the bounded constraint $D(\cdot) \in [-1, 1]$, the optimal $D^*$ maximizing the above summation becomes (Nowozin et al., 2016):

$$D^*(x) = \mathsf{sign}\{\mathbb{P}_{\mathsf{bias}}(x) - \mathbb{P}_G(x)\}.$$

Substituting this into equation 8, we obtain:

$$
\begin{aligned}
&\min_G \ \sum_{x \in \mathcal{X}} \big\{\mathbb{P}_{\mathsf{bias}}(x) - \mathbb{P}_G(x)\big\} D^*(x) \\
&= \min_G \ \sum_{x \in \mathcal{X}} |\mathbb{P}_{\mathsf{bias}}(x) - \mathbb{P}_G(x)| = \min_G \ 2 \cdot \mathsf{TV}(\mathbb{P}_{\mathsf{bias}}, \mathbb{P}_G)
\end{aligned}
$$

where the first equality is due to the optimal $D^*$. This completes the proof. □

## B   APPENDIX: IMPLEMENTATION DETAILS

We first present $m_{\mathsf{bias}}$ (the number of samples for training set) and $m_{\mathsf{ref}}$ (w.r.t. reference set) employed in a variety of considered scenarios. Next we provide details on the model architectures (together with hyperparameters) employed for $(G(\cdot), D(\cdot), D_{\mathsf{fair}}(\cdot))$ (in our framework) as well as for attribute classifiers.

### B.1   THE NUMBER OF SAMPLES FOR TRAINING AND REFERENCE SETS

Tables 4 and 5 show $m_{\mathsf{bias}}$ and $m_{\mathsf{ref}}$ employed for our experiments. Notice that $m_{\mathsf{bias}}$ (or $m_{\mathsf{ref}}$) in UTKFace and FairFace is much smaller than that of CelebA. As expected, this leads to the overall performance degradation of UTKFace and FairFace, as reported in appendix E; see Tables, 15, 16, and 17 therein.

Table 4: Number of samples used in CelebA. CelebA-single indicates the scenario in which a single attribute (gender) is employed. CelebA-multi refers to the case considering two attributes: gender and hair color. For CelebA-single, we consider $9 : 1$ female-vs-male samples for training set. For CelebA-multi, we consider $85 : 15$ non-black-hair vs black-hair samples. The reference set size indicates a ratio relative to training set.

| | CelebA-single | | | | | CelebA-multi | |
|---|---|---|---|---|---|---|---|
| Reference set size | $1\%$ | $2.5\%$ | $5\%$ | $10\%$ | $25\%$ | $10\%$ | $25\%$ |
| $m_{\mathsf{bias}}$ | $67,507$ | $67,507$ | $67,507$ | $67,507$ | $67,507$ | $60,000$ | $60,000$ |
| $m_{\mathsf{ref}}$ | $674$ | $1,686$ | $3,374$ | $6,750$ | $16,876$ | $6,000$ | $15,000$ |

Table 5: Number of samples used in UTKFace and FairFace. All the settings are the same as those in Table 4. For UTKFace, we consider $9 : 1$ white-vs-non-white samples for training set. For FairFace, we consider $9 : 1$ ratio white-vs-black samples.

|  | UTKFace | | FairFace | |
|---|---|---|---|---|
| Reference set size | $10\%$ | $25\%$ | $10\%$ | $25\%$ |
| $m_{\text{bias}}$ | $8,486$ | $7,865$ | $17,397$ | $16,123$ |
| $m_{\text{ref}}$ | $848$ | $1,966$ | $1,740$ | $4,032$ |

### B.2 ARCHITECTURES AND HYPERPARAMETER CONFIGURATIONS

We employ attribute classifiers, only for the purpose of evaluating our twin measures: (i) fairness discrepancy (defined in equation 1); (ii) intra FID. We introduce four different attribute classifiers for predicting sensitive attributes in the following scenarios: (i) gender for CelebA-single; (ii) gender and hair-color for CelebA-multi; (iii) white-vs-non-white race for UTKFace; (iv) white-vs-black race for FairFace. For all the attribute classifiers, we use a modified version of ResNet18 (He et al., 2016); see Table 6 for architectural details. We train the classifiers for CelebA and FairFace scenarios with the standard train and validation splits of CelebA (Liu et al., 2015) and FairFace (Karkkainen & Joo, 2021), respectively. UTKFace (Zhang et al., 2017) does not provide such standard dataset splits, so we manually divide the full UTKFace dataset into $8 : 1 : 1$ ratio of train/valid/test splits and use them for training the attribute classifier for UTKFace. We use the same hyperparameters for training all of the classifiers. We use a batch size of $64$ and adam optimizer with the learning rate $0.001$ and $(\beta_1, \beta_2) = (0.9, 0.999)$. We also employ early stopping.

We adopt the BigGAN (Brock et al., 2019) architecture for implementing all the baselines (Baselines I, II, and Choi et al. (2020)) and the proposed framework. See Table 7 for architectural details. For $D_{\text{fair}}$ employed in our framework, we use the same architecture as $D$. Tables 8, 9, and 10 exhibit hyperparameters used for our approach. Tables 11 and 12 exhibit hyperparameters used for experiments with other regularizations. For implementing JSD-based regularization, we rely upon the original GAN with non-saturating loss (Goodfellow et al., 2014). For Kullback-Leibler divergence, we employ the loss function proposed in $f$-GAN (Nowozin et al., 2016). Least Squares GAN loss (Mao et al., 2017) is employed for implementation of Pearson-$\chi^2$ divergence. For implementing Wasserstein distance, WGAN-GP (Gradient Penalty) (Gulrajani et al., 2017) is employed.

For all the experimental results, $\lambda$ was chosen as the one that yields the lowest FID among all the candidates respecting a good fairness performance, e.g., fairness discrepancy $< 0.1$ on CelebA dataset. For a batch size, we considered the following candidates $\{8, 16, 32, 64, 128, 256\}$. We controlled the learning rates for $G$ and $D$ among $\{5 \cdot 10^{-5}, 10^{-4}, 2 \cdot 10^{-4}, 4 \cdot 10^{-4}\}$ and $\{2 \cdot 10^{-4}, 4 \cdot 10^{-4}\}$ respectively. For the number of $D$-step per $G$-step, we considered a number among $\{2, 3, 4\}$.

Table 6: A modified ResNet18 architecture for attribute classifiers.

| Name | Component |
|---|---|
| Conv | $7 \times 7$ conv, 64 filters, stride 2 |
| Residual Block 1 | $3 \times 3$ max pool, stride 2 |
| Residual Block 2 | $\begin{bmatrix} 3 \times 3 \text{ conv, 128 filters} \\ 3 \times 3 \text{ conv, 128 filters} \end{bmatrix} \times 2$ |
| Residual Block 3 | $\begin{bmatrix} 3 \times 3 \text{ conv, 256 filters} \\ 3 \times 3 \text{ conv, 256 filters} \end{bmatrix} \times 2$ |
| Residual Block 4 | $\begin{bmatrix} 3 \times 3 \text{ conv, 512 filters} \\ 3 \times 3 \text{ conv, 512 filters} \end{bmatrix} \times 2$ |
| Output Layer | $7 \times 7$ average pool stride 1, fully-connected, softmax |

Table 7: BigGAN architecture for generator and discriminator. Here $ch$ represents the channel width multiplier: $ch = 64$ for $64 \times 64$ images of our interest. "ResBlock up" means a Generator Residual Block where the input is passed through a ReLU activation and then upsampled, followed by two $3 \times 3$ convolutional layers with another ReLU activation in between. "ResBlock down" refers to a Discriminator Residual Block in which the input is passed through two $3 \times 3$ convolutional layers with a ReLU activation in between, and then downsampled. Upsampling is implemented via nearest neighbor interpolation, and downsampling is via average pooling. "ResBlock up/down $n \to m$" refers to a ResBlock with $n$ input channels and $m$ output channels.

| $G$ | $D$ (or $D_{\mathsf{fair}}$) |
|---|---|
| $1 \times 1 \times 2ch$ Noise | $64 \times 64 \times 3$ Image |
| Linear $1 \times 1 \times 16ch \to 1 \times 1 \times 16ch$
ResBlock up $16ch \to 16ch$ | ResBlock down $1ch \to 2ch$
Non-Local Block $(64 \times 64)$ |
| ResBlock up $16ch \to 8ch$ | ResBlock down $2ch \to 4ch$ |
| ResBlock up $8ch \to 4ch$ | ResBlock down $4ch \to 8ch$ |
| ResBlock up $4ch \to 2ch$ | ResBlock down $8ch \to 16ch$ |
| Non-Local Block $(64 \times 64)$
ResBlock up $4ch \to 2ch$ | ResBlock down $16ch \to 16ch$
ResBlock $16ch \to 16ch$ |
| BN, ReLU, $3 \times 3$ Conv $1ch \to 3$ | ReLU, Global sum pooling |
| Tanh | Linear $\to 1$ |

For the second discriminator w.r.t fairness, we swept the learning rate through $\{10^{-4}, 2 \cdot 10^{-4}, 4 \cdot 10^{-4}\}$ and the number of $D$-step per $G$-step through $\{1, 2, 3\}$. For Choi et al. (2020), we found for UTKFace and FairFace that a simple 3 or 5-layer CNN classifier better estimates example weights than the ResNet-based architecture used in Choi et al. (2020). All baseline performances are measured using the same approach as in Choi et al. (2020): average over 10 independent evaluation sets of 10,000 samples each drawn from the corresponding baseline model.

Table 8: Hyperparameters used for CelebA experiments. We use the same betas for $(G, D, D_{\mathsf{fair}})$.

|  | CelebA-single | | CelebA-multi | |
|---|---|---|---|---|
| Reference set size | 10% | 25% | 10% | 25% |
| $\lambda$ | 0.9 | 0.85 | 0.9 | 0.925 |
| Batch size ($D$ & $G$) | 16 | 8 | 16 | 16 |
| Batch size ($D_{\mathsf{fair}}$) | 8 | 8 | 8 | 16 |
| Learning rate ($G$) | 5e-5 | 5e-5 | 5e-5 | 5e-5 |
| Learning rate ($D$ & $D_{\mathsf{fair}}$) | 2e-4 | 2e-4 | 2e-4 | 2e-4 |
| $(\beta_1, \beta_2)$ (for Adam optimizer) | $(0, 0.999)$ | $(0, 0.999)$ | $(0, 0.999)$ | $(0, 0.999)$ |
| $D$ steps per $G$ step | 2 | 2 | 2 | 2 |
| $D_{\mathsf{fair}}$ steps per $G$ step | 1 | 1 | 1 | 1 |

## C APPENDIX: ADDITIONAL DISCUSSIONS

We first investigate the evaluation sensitivity w.r.t. the attribute classifiers that we employ for performance evaluation. Next, we discuss the impact of the reference set size on the fairness performance in our framework. Lastly, we explore a connection between our interested setting and transfer learning framework.

Table 9: Hyperparameters used for CelebA experiments with small reference set sizes. We share $(\beta_1, \beta_2) = (0, 0.999)$ for $(G, D, D_{\mathsf{fair}})$.

| | CelebA-single | | |
|---|---|---|---|
| Reference set size | 1% | 2.5% | 5% |
| $\lambda$ | 0.9 | 0.9 | 0.9 |
| Batch size ($D$ & $G$) | 8 | 16 | 16 |
| Batch size ($D_{\mathsf{fair}}$) | 4 | 8 | 8 |
| Learning rate ($G$) | 5e-5 | 5e-5 | 5e-5 |
| Learning rate ($D$ & $D_{\mathsf{fair}}$) | 2e-4 | 2e-4 | 2e-4 |
| $(\beta_1, \beta_2)$ (for Adam optimizer) | $(0, 0.999)$ | $(0, 0.999)$ | $(0, 0.999)$ |
| $D$ steps per $G$ step | 2 | 2 | 2 |
| $D_{\mathsf{fair}}$ steps per $G$ step | 1 | 1 | 1 |

Table 10: Hyperparameters used for UTKFace and FairFace experiments. We employ the same betas for $(G, D, D_{\mathsf{fair}})$.

| | UTKFace | | FairFace | |
|---|---|---|---|---|
| Reference set size | 10% | 25% | 10% | 25% |
| $\lambda$ | 0.9 | 0.8 | 0.9 | 0.85 |
| Batch size ($D$ & $G$) | 16 | 16 | 16 | 16 |
| Batch size ($D_{\mathsf{fair}}$) | 8 | 8 | 8 | 8 |
| Learning rate ($G$) | 5e-5 | 5e-5 | 5e-5 | 5e-5 |
| Learning rate ($D$ & $D_{\mathsf{fair}}$) | 2e-4 | 2e-4 | 2e-4 | 2e-4 |
| $(\beta_1, \beta_2)$ (for Adam optimizer) | $(0, 0.999)$ | $(0, 0.999)$ | $(0, 0.999)$ | $(0, 0.999)$ |
| $D$ steps per $G$ step | 2 | 2 | 2 | 2 |
| $D_{\mathsf{fair}}$ steps per $G$ step | 1 | 1 | 1 | 1 |

Table 11: Hyperparameters used for experiments with other fairness regularizers. We employ the same betas for $(G, D, D_{\mathsf{fair}})$. "JSD-based" refers to a regularization method based on Jensen-Shannon divergence (Wong & You, 1985) implemented via Goodfellow et al. (2014). "KLD-based" is the one built upon Kullback-Leibler divergence (Kullback & Leibler, 1951; Nowozin et al., 2016). "$\chi^2$-based" represents the one implementing Pearson-$\chi^2$ divergence (Pearson, 1900; Mao et al., 2017). "WD-based" refers to a regularization with Wasserstein distance (Wasserstein, 1969; Gulrajani et al., 2017).

| | JSD-based | KLD-based | $\chi^2$-based | WD-based |
|---|---|---|---|---|
| $\lambda$ | 0.985 | 0.925 | 0.9 | 0.975 |
| Batch size ($D$ & $G$) | 16 | 16 | 16 | 16 |
| Batch size ($D_{\mathsf{fair}}$) | 8 | 8 | 8 | 8 |
| Learning rate ($G$) | 5e-5 | 5e-5 | 5e-5 | 1e-4 |
| Learning rate ($D$ & $D_{\mathsf{fair}}$) | 2e-4 | 2e-4 | 2e-4 | 1e-4 |
| $(\beta_1, \beta_2)$ (for Adam optimizer) | $(0, 0.999)$ | $(0, 0.999)$ | $(0, 0.999)$ | $(0, 0.9)$ |
| $D$ steps per $G$ step | 2 | 2 | 2 | 7 |
| $D_{\mathsf{fair}}$ steps per $G$ step | 1 | 2 | 1 | 7 |

## C.1 EVALUATION SENSITIVITY W.R.T. ATTRIBUTE CLASSIFIERS

As mentioned in Sections 2 and 4.1, we rely upon the attribute classifier for evaluating our two performance measures, *fairness discrepancy* (see Definition 1 in Section 2) and intra FID. One natural question is then to ask how sensitive the evaluation is to the choice of the attribute classifier. We address this question by investigating a gap between performances measured with two differently-accurate classifiers. We found that fairness discrepancy varies significantly depending on the accuracy of the attribute classifier. This is in contrast with intra FID which is not much affected with the choice

Table 12: Hyperparameters used for experiments with other fairness regularizers. "JSD-JSD" indicates the case where both base optimization (the first term in equation 5) and regularization are based on Jensen-Shannon divergence (Wong & You, 1985). "$\chi^2$-$\chi^2$" refers to the one that implements Pearson-$\chi^2$ divergence (Pearson, 1900) for base optimization as well as for regularization. "WD-WD" represents the one that employs Wasserstein distance (Wasserstein, 1969) both for base optimization and regularization.

|  | JSD-JSD | $\chi^2$-$\chi^2$ | WD-WD |
|---|---|---|---|
| $\lambda$ | 0.99 | 0.75 | 0.975 |
| Batch size $(D\ \&\ G)$ | 16 | 16 | 64 |
| Batch size $(D_{\text{fair}})$ | 8 | 8 | 16 |
| Learning rate $(G)$ | 5e-5 | 5e-5 | 1e-4 |
| Learning rate $(D\ \&\ D_{\text{fair}})$ | 2e-4 | 2e-4 | 1e-4 |
| $(\beta_1, \beta_2)$ (for Adam optimizer) | $(0, 0.999)$ | $(0, 0.999)$ | $(0, 0.9)$ |
| $D$ steps per $G$ step | 2 | 2 | 7 |
| $D_{\text{fair}}$ steps per $G$ step | 1 | 1 | 7 |

of the attribute classifier. For instance, 10% accuracy of the attribute classifier yields around 25% deviations in fairness discrepancy, while exhibiting marginal deviations in intra FID values less than 1%.

## C.2 IMPACT OF REFERENCE SET SIZE ON FAIRNESS PERFORMANCE

Our framework heavily depends on reference data in regulating unfairness, and the size of the reference data indeed affects our performance of fairness. If a given reference dataset is too small to well-represent the underlying reference distribution, the measured TVD (via such small-sized reference set) would then be inaccurate, degrading the fairness performance accordingly.

On the other hand, such degradation can occur with a large reference dataset. For implementing the fairness regularization, we rely upon GAN framework that often yields a model focused on learning only a specific part of representations in a given dataset, rather than covering all the representations inside (Metz et al., 2016; Arjovsky et al., 2017). Due to this property, with a larger reference set augmented with vast amount of representations, our models (generator and fairness discriminator) can be more encouraged toward learning such augmented (yet less relevant to representations w.r.t. demographic groups) representations and therefore less focused on respecting balanced representations of demographic groups, degrading the fairness performance accordingly. This phenomenon is well exhibited in our experiments; see the last rows in Tables 1 and 15 for instance.

## C.3 RELATION TO TRANSFER LEARNING

Our problem setting can be interpreted as a transfer learning framework, where we wish to exploit the knowledge learned from a large training dataset $\mathcal{D}_{\text{bias}}$ to augment learning on a small reference dataset $\mathcal{D}_{\text{ref}}$. We explore such connection herein with a basic transfer learning technique, fine-tuning. Employing only one discriminator, we first train our models on $\mathcal{D}_{\text{bias}}$ until convergence, and then fine-tune the models on $\mathcal{D}_{\text{ref}}$. Table 13 compares the performance of the fine-tuning method with all the other approaches, evaluated on CelebA-single with 10% reference set. Notice that the fine-tuning method well-balances between Baseline I and II, yet still inferior than the proposed approach both in fairness and sample quality.

Table 13: Performance comparison with fine-tuning method on CelebA-single with 10% reference set. Baseline I is a non-fair algorithm building upon the base framework with the hinge loss (Lim & Ye, 2017; Tran et al., 2017), and trained with the aggregated data $\mathcal{D}_{bias} \cup \mathcal{D}_{ref}$. Baseline II is the same non-fair algorithm yet trained only with the small yet balanced reference dataset $\mathcal{D}_{ref}$. Choi et al. is the state of the art (Choi et al., 2020). "Female" (or "Male") refers to intra FID for female (or male) group. The lower intra FID, the more realistic and diverse samples. "Fairness" is *fairness discrepancy* introduced by Choi et al. (2020); see equation 1 for the definition. The lower, the fairer.

|  | Female | Male | Fairness |
| --- | --- | --- | --- |
| Baseline I | $12.73 \pm 0.053$ | $8.37 \pm 0.057$ | $0.554 \pm 0.002$ |
| Baseline II | $32.31 \pm 0.109$ | $26.18 \pm 0.049$ | $0.115 \pm 0.002$ |
| Choi et al. | $25.74 \pm 0.079$ | $21.07 \pm 0.064$ | $0.104 \pm 0.002$ |
| Fine-tuning | $19.79 \pm 0.451$ | $13.40 \pm 0.514$ | $0.147 \pm 0.008$ |
| Proposed | $14.29 \pm 1.354$ | $9.51 \pm 1.069$ | $0.057 \pm 0.012$ |

## D  APPENDIX: COMPLEXITY ANALYSIS

We also offer complexity analysis of our approach while making a comparison to Choi et al. (2020). As performance measures, we consider the total number of parameters as well as the running time measured under Pytorch on Xeon Silver 4210R CPU and TITAN RTX GPU. Table 14 presents the interested measures (together with fairness and intra FID performances) for one certain scenario: CelebA-single with 10% reference set size. While our approach provides better intra FID and fairness performances, it comes at a cost of an increased complexity, around 16.7% and 40% for the running time and model complexity, respectively.

Table 14: Performance comparison on CelebA-single with 10% reference set size. Choi et al. (2020) corresponds to the state of the art. "Intra FID" refers to Fréchet Inception Distance (Heusel et al., 2017) computed within each group (Miyato & Koyama, 2018; Zhang et al., 2019), and we provide results for minority group herein. The lower intra FID, the more realistic and diverse samples. "Fairness" is fairness discrepancy introduced by Choi et al. (2020). The lower, the fairer.

|  | Intra FID | Fairness | Running Time (s) | # of parameters |
| --- | --- | --- | --- | --- |
| Choi et al. | $25.74 \pm 0.079$ | $0.104 \pm 0.002$ | $42,417.24$ | $51,577,028$ |
| Proposed | $14.29 \pm 1.354$ | $0.057 \pm 0.012$ | $49,497.72$ | $71,120,773$ |

## E  APPENDIX: ADDITIONAL RESULTS

We first provide performance comparison with the three baselines (Baseline I, Baseline II, and Choi et al. (2020)) on UTKFace and FairFace datasets. Next we present the complete results on CelebA dataset, where intra FID values for all demographics are included. We then compare fairness regularization performances with distinct base optimizations. Lastly, we provide tradeoff curves that illustrate the tension between fairness and sample quality in our framework.

### E.1  PERFORMANCES ON UTKFACE AND FAIRFACE

Tables 15 and 16 concern UTKFace and FairFace datasets. One significant distinction w.r.t. CelebA is that training and reference set sizes are much smaller, around 7 times. Hence, as expected, the overall performances are worse than those on CelebA dataset. Even in this small data regime, we observe the same trends on the performance benefits of ours relative to the baselines.

Table 15: Performance comparison on UTKFace dataset where a race attribute is considered: white vs. non-white. Baseline I is a non-fair algorithm building upon the base framework with the hinge loss (Lim & Ye, 2017; Tran et al., 2017), and trained with the aggregated data $\mathcal{D}_{\mathsf{bias}} \cup \mathcal{D}_{\mathsf{ref}}$. Baseline II is the same non-fair algorithm yet trained only with the small yet balanced reference dataset $\mathcal{D}_{\mathsf{ref}}$. Choi et al. is the state of the art (Choi et al., 2020). "White" (or "Non-white") refers to intra FID (Miyato & Koyama, 2018; Zhang et al., 2019) for white (or non-white) group. The lower intra FID, the more realistic and diverse samples for that group. "Fairness" is *fairness discrepancy* introduced by Choi et al. (2020); see equation 1 for the definition. The lower, the fairer. The reference set size indicates a ratio relative to training data.

| Reference set size | | 10% | 25% |
|---|---|---|---|
| Baseline I | White | $15.37 \pm 0.053$ | $14.17 \pm 0.047$ |
| | Non-white | $19.89 \pm 0.119$ | $18.86 \pm 0.117$ |
| | Fairness | $0.453 \pm 0.002$ | $0.400 \pm 0.003$ |
| Baseline II | White | $77.83 \pm 0.136$ | $36.78 \pm 0.070$ |
| | Non-white | $83.51 \pm 0.071$ | $35.73 \pm 0.077$ |
| | Fairness | $0.010 \pm 0.003$ | $0.007 \pm 0.003$ |
| Choi et al. | White | $28.07 \pm 0.068$ | $33.95 \pm 0.081$ |
| | Non-white | $36.43 \pm 0.231$ | $40.06 \pm 0.138$ |
| | Fairness | $0.285 \pm 0.003$ | $0.123 \pm 0.003$ |
| Proposed | White | $23.64 \pm 3.153$ | $16.69 \pm 0.913$ |
| | Non-white | $27.24 \pm 4.125$ | $19.20 \pm 2.285$ |
| | Fairness | $0.091 \pm 0.022$ | $0.131 \pm 0.021$ |

Table 16: Performance comparison on FairFace dataset. All the settings and baselines are the same as those in Table 15, except for different types of demographics: white vs. black.

| Reference set size | | 10% | 25% |
|---|---|---|---|
| Baseline I | White | $25.38 \pm 0.084$ | $23.68 \pm 0.109$ |
| | Black | $25.76 \pm 0.068$ | $22.96 \pm 0.047$ |
| | Fairness | $0.434 \pm 0.002$ | $0.386 \pm 0.003$ |
| Baseline II | White | $77.45 \pm 0.104$ | $48.02 \pm 0.117$ |
| | Black | $83.76 \pm 0.118$ | $45.20 \pm 0.055$ |
| | Fairness | $0.105 \pm 0.002$ | $0.009 \pm 0.002$ |
| Choi et al. | White | $33.20 \pm 0.059$ | $36.56 \pm 0.106$ |
| | Black | $33.33 \pm 0.076$ | $36.37 \pm 0.082$ |
| | Fairness | $0.317 \pm 0.002$ | $0.142 \pm 0.002$ |
| Proposed | White | $30.73 \pm 2.210$ | $28.67 \pm 2.330$ |
| | Black | $31.16 \pm 2.158$ | $34.07 \pm 3.140$ |
| | Fairness | $0.044 \pm 0.019$ | $0.106 \pm 0.023$ |

E.2 COMPLETE RESULTS ON CELEBA

Tables 17 to 19 exhibit the complete results w.r.t. CelebA experiments, where intra FID values for all associated groups are provided. Observe that in all considered settings, our approach offers better (or comparable) sample quality than the state of the art (Choi et al., 2020) for *all* demographics, reflected in low intra FID values.

Table 17: Complete results on CelebA-single in which a *single* attribute (gender) is employed. Baseline I is a non-fair algorithm building upon the base framework with the hinge loss (Lim & Ye, 2017; Tran et al., 2017), and trained with the aggregated data $\mathcal{D}_{\text{bias}} \cup \mathcal{D}_{\text{ref}}$. Baseline II is the same non-fair algorithm yet trained only with the small yet balanced reference dataset $\mathcal{D}_{\text{ref}}$. Choi et al. (2020) is the state of the art. "Female" (or "Male") refers to intra FID (Miyato & Koyama, 2018; Zhang et al., 2019) for female (or male) group. The lower intra FID, the more realistic and diverse samples for that group. "Fairness" is *fairness discrepancy* introduced by Choi et al. (2020); see equation 1 for the definition. The lower, the fairer. The reference set size indicates a ratio relative to training data.

| Reference set size | | 10% | 25% |
|---|---|---|---|
| Baseline I | Female | $8.37 \pm 0.057$ | $7.32 \pm 0.047$ |
| | Male | $12.73 \pm 0.053$ | $12.00 \pm 0.069$ |
| | Fairness | $0.554 \pm 0.002$ | $0.495 \pm 0.001$ |
| Baseline II | Female | $26.18 \pm 0.049$ | $17.66 \pm 0.084$ |
| | Male | $32.31 \pm 0.109$ | $23.81 \pm 0.118$ |
| | Fairness | $0.115 \pm 0.002$ | $0.093 \pm 0.002$ |
| Choi et al. | Female | $21.07 \pm 0.064$ | $21.33 \pm 0.059$ |
| | Male | $25.74 \pm 0.079$ | $20.68 \pm 0.076$ |
| | Fairness | $0.104 \pm 0.002$ | $0.065 \pm 0.002$ |
| Proposed | Female | $9.51 \pm 1.069$ | $7.45 \pm 1.351$ |
| | Male | $14.29 \pm 1.354$ | $11.34 \pm 1.288$ |
| | Fairness | $0.057 \pm 0.012$ | $0.069 \pm 0.015$ |

Table 18: Complete results on CelebA-multi experiments where two attributes are considered: gender and hair color. All the settings and baselines are the same as those in Table 17 yet here we consider intra FID values for four groups: (i) female with non-black hair; (ii) male with non-black hair; (iii) female with black-hair; (iv) male with black hair.

| Reference set size | | 10% | 25% |
|---|---|---|---|
| Baseline I | Female w/ non-black hair | $9.25 \pm 0.040$ | $7.74 \pm 0.082$ |
| | Male w/ non-black hair | $12.70 \pm 0.090$ | $11.61 \pm 0.065$ |
| | Female w/ black hair | $8.93 \pm 0.049$ | $8.14 \pm 0.043$ |
| | Male w/ black hair | $10.83 \pm 0.046$ | $9.76 \pm 0.067$ |
| | Fairness | $0.245 \pm 0.002$ | $0.224 \pm 0.002$ |
| Baseline II | Female w/ non-black hair | $26.96 \pm 0.065$ | $19.22 \pm 0.087$ |
| | Male w/ non-black hair | $35.35 \pm 0.116$ | $25.73 \pm 0.131$ |
| | Female w/ black hair | $24.51 \pm 0.085$ | $17.15 \pm 0.078$ |
| | Male w/ black hair | $29.66 \pm 0.088$ | $21.92 \pm 0.056$ |
| | Fairness | $0.136 \pm 0.001$ | $0.107 \pm 0.001$ |
| Choi et al. | Female w/ non-black hair | $15.86 \pm 0.048$ | $13.11 \pm 0.032$ |
| | Male w/ non-black hair | $18.78 \pm 0.058$ | $14.78 \pm 0.058$ |
| | Female w/ black hair | $16.88 \pm 0.047$ | $12.85 \pm 0.039$ |
| | Male w/ black hair | $16.27 \pm 0.042$ | $13.68 \pm 0.062$ |
| | Fairness | $0.090 \pm 0.001$ | $0.063 \pm 0.002$ |
| Proposed | Female w/ non-black hair | $9.14 \pm 0.733$ | $8.84 \pm 0.457$ |
| | Male w/ non-black hair | $11.93 \pm 0.656$ | $11.82 \pm 0.451$ |
| | Female w/ black hair | $11.01 \pm 0.390$ | $9.75 \pm 0.435$ |
| | Male w/ black hair | $12.93 \pm 0.631$ | $12.07 \pm 0.299$ |
| | Fairness | $0.080 \pm 0.006$ | $0.073 \pm 0.002$ |

Table 19: Complete results on CelebA-single with the reference set size down to $1\%$. All the other settings and baselines are the same as those in Table 17.

| Reference set size | | 5% | 2.5% | 1% |
|---|---|---|---|---|
| Baseline I | Female | $8.25 \pm 0.039$ | $8.09 \pm 0.064$ | $8.31 \pm 0.057$ |
| | Male | $13.54 \pm 0.074$ | $13.79 \pm 0.072$ | $15.89 \pm 0.094$ |
| | Fairness | $0.559 \pm 0.001$ | $0.566 \pm 0.002$ | $0.576 \pm 0.002$ |
| Baseline II | Female | $33.56 \pm 0.036$ | $52.46 \pm 0.070$ | $83.46 \pm 0.100$ |
| | Male | $40.07 \pm 0.062$ | $67.70 \pm 0.112$ | $92.34 \pm 0.131$ |
| | Fairness | $0.120 \pm 0.003$ | $0.018 \pm 0.003$ | $0.455 \pm 0.002$ |
| Choi et al. | Female | $25.51 \pm 0.044$ | $16.94 \pm 0.046$ | $21.49 \pm 0.057$ |
| | Male | $32.30 \pm 0.049$ | $25.81 \pm 0.043$ | $30.28 \pm 0.085$ |
| | Fairness | $0.043 \pm 0.002$ | $0.274 \pm 0.002$ | $0.355 \pm 0.002$ |
| Proposed | Female | $15.04 \pm 0.635$ | $17.03 \pm 1.174$ | $21.88 \pm 1.946$ |
| | Male | $23.42 \pm 1.312$ | $24.53 \pm 1.157$ | $30.93 \pm 2.196$ |
| | Fairness | $0.048 \pm 0.012$ | $0.119 \pm 0.011$ | $0.166 \pm 0.011$ |

### E.3    REGULARIZATION WITH DIFFERENT BASE OPTIMIZATIONS

Table 20 compares regularization performances with different divergence measures for the first term in equation 5. Even in this comparison, we observe that our TVD-based approach is still prominent among the considered methods, yielding well-balanced performances of fairness and sample quality. We do not include the performance regarding Kullback-Leibler divergence, where we failed to get meaningful results due to numerical problem. This is also reported in other works (Song & Ermon, 2020) which relies upon the same implementation (Nowozin et al., 2016).

Table 20: Performance comparison with other fairness regularizations on CelebA-single with the 10% reference set size. "JSD-JSD" indicates the case where both base optimization (the first term in equation 5) and regularization are based on Jensen-Shannon divergence (Wong & You, 1985; Goodfellow et al., 2014). "$\chi^2$-$\chi^2$" refers to the one that implements Pearson-$\chi^2$ divergence (Pearson, 1900; Mao et al., 2017) for base optimization as well as for regularization. "WD-WD" represents the one that employs Wasserstein distance (Wasserstein, 1969; Gulrajani et al., 2017) both for base optimization and regularization. "Female" (or "Male") refers to intra FID for female (or male) group. For each measure, we mark the best result in bold and the second-best with underline.

|          | JSD-JSD | $\chi^2$-$\chi^2$ | WD-WD | TVD-TVD |
|----------|---------|---------|---------|---------|
| Female   | $\underline{11.02 \pm 1.521}$ | $26.68 \pm 1.496$ | $15.80 \pm 1.049$ | $\mathbf{9.51 \pm 1.069}$ |
| Male     | $\underline{15.33 \pm 1.661}$ | $40.35 \pm 2.605$ | $21.86 \pm 0.782$ | $\mathbf{14.29 \pm 1.354}$ |
| Fairness | $0.107 \pm 0.015$ | $\underline{0.054 \pm 0.012}$ | $\mathbf{0.037 \pm 0.014}$ | $0.057 \pm 0.012$ |

### E.4    FAIRNESS-QUALITY TRADEOFF CURVES

Fig. 4 visualizes a tradeoff relationship between sample quality and fairness in our framework. Despite slight fluctuations, one can readily see that as lambda increases, fairness performance improves (smaller fairness discrepancy) at the expense of the degraded sample quality, reflected in larger intra FID. This validates the role of $\lambda$ as a tuning knob for controlling fairness.

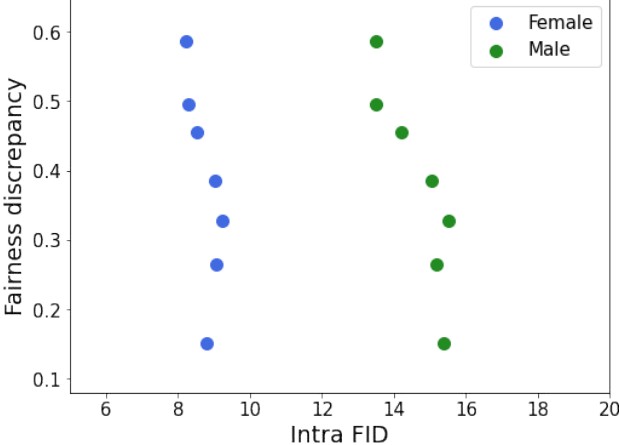

Figure 4: Fairness-quality tradeoff curves evaluated on CelebA-single with 10% reference set. Each point is obtained with a particular $\lambda$, the tuning knob in our framework. Blue dot points indicate performances for female group, and green dots are for male group.

## F  APPENDIX: ADDITIONAL GENERATED SAMPLES

We provide generated samples obtained from many scenarios under the three real datasets, not included in the main paper due to space constraint. Unlike the main paper, we also present samples generated from Baseline I and Baseline II. Remember that Baseline I is a non-fair algorithm building upon the base framework (equation 2 in Section 2) with the hinge loss (Lim & Ye, 2017; Tran et al., 2017), while being trained with the aggregated dataset $\mathcal{D}_{\mathsf{bias}} \cup \mathcal{D}_{\mathsf{ref}}$. Baseline II is the same non-fair algorithm yet trained only with $\mathcal{D}_{\mathsf{ref}}$. We leave all the samples from below.

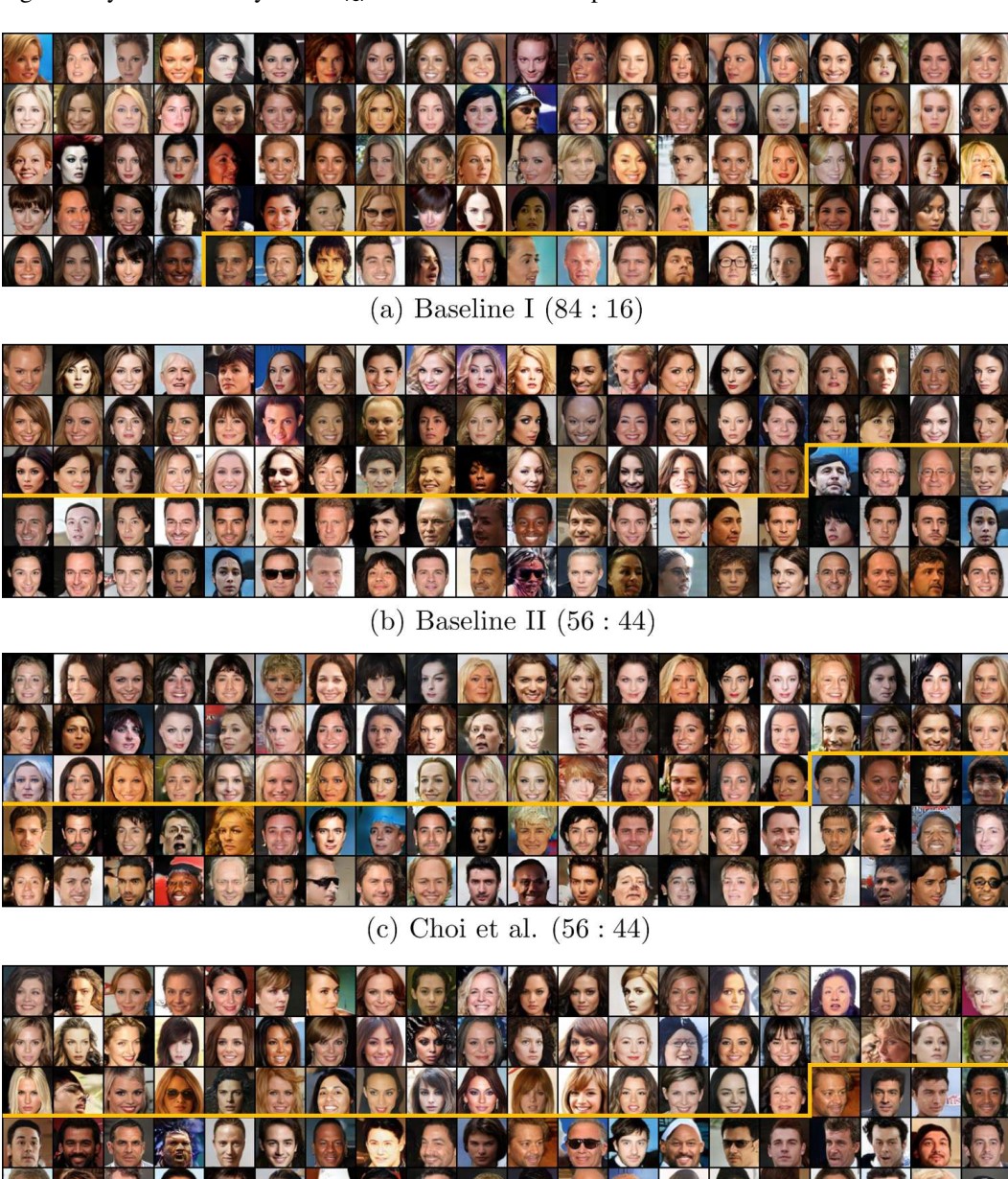

(a) Baseline I (84 : 16)

(b) Baseline II (56 : 44)

(c) Choi et al. (56 : 44)

(d) proposed (56 : 44)

Figure 5: Generated samples trained on CelebA-single with 25% reference set size. Faces above the yellow line are female, while the rest are male.

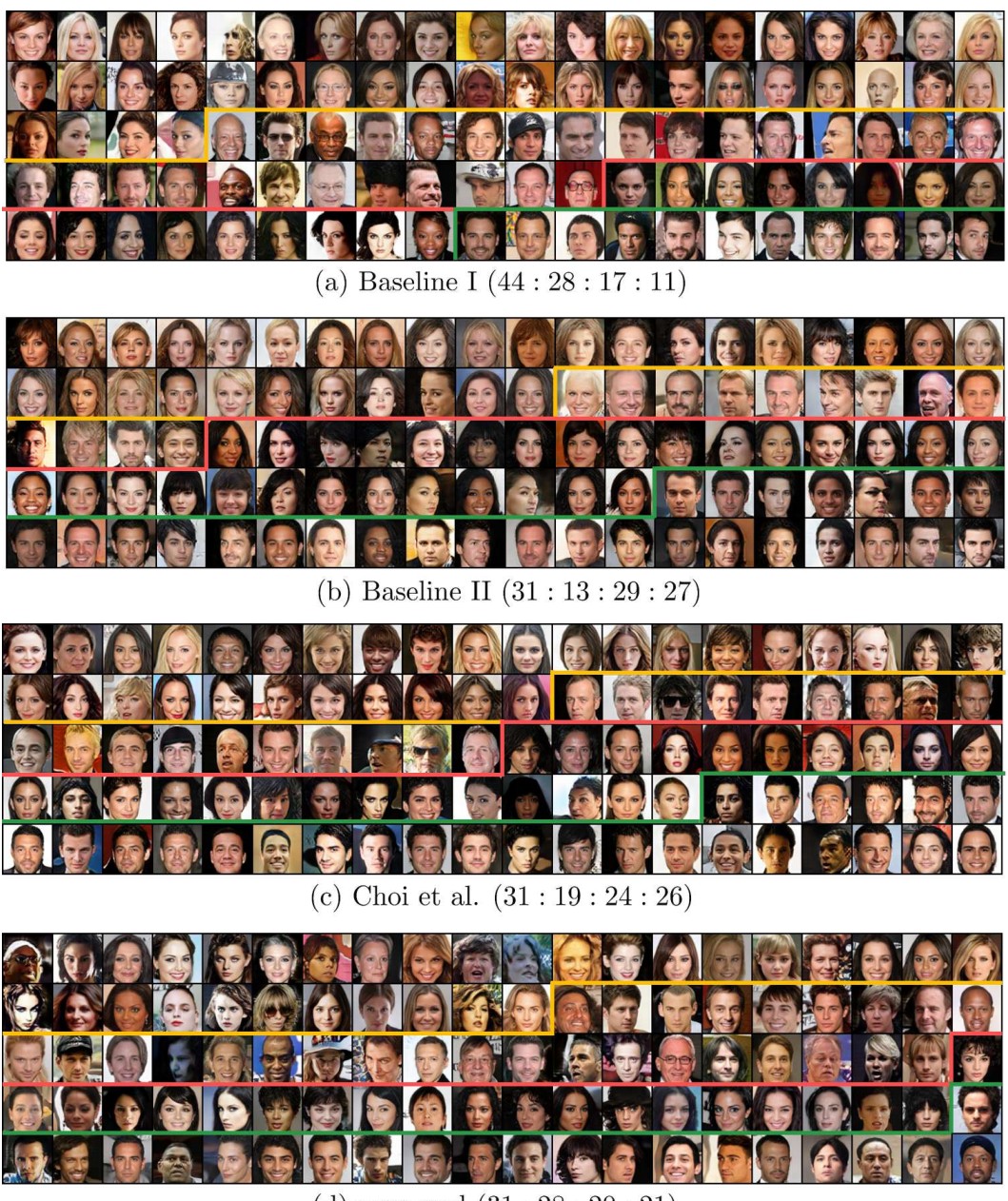

(a) Baseline I (44 : 28 : 17 : 11)

(b) Baseline II (31 : 13 : 29 : 27)

(c) Choi et al. (31 : 19 : 24 : 26)

(d) proposed (31 : 28 : 20 : 21)

Figure 6: Generated samples trained on CelebA-multi with 10% reference set size. Faces above the red line are samples without black hair, while the rest are black hair.

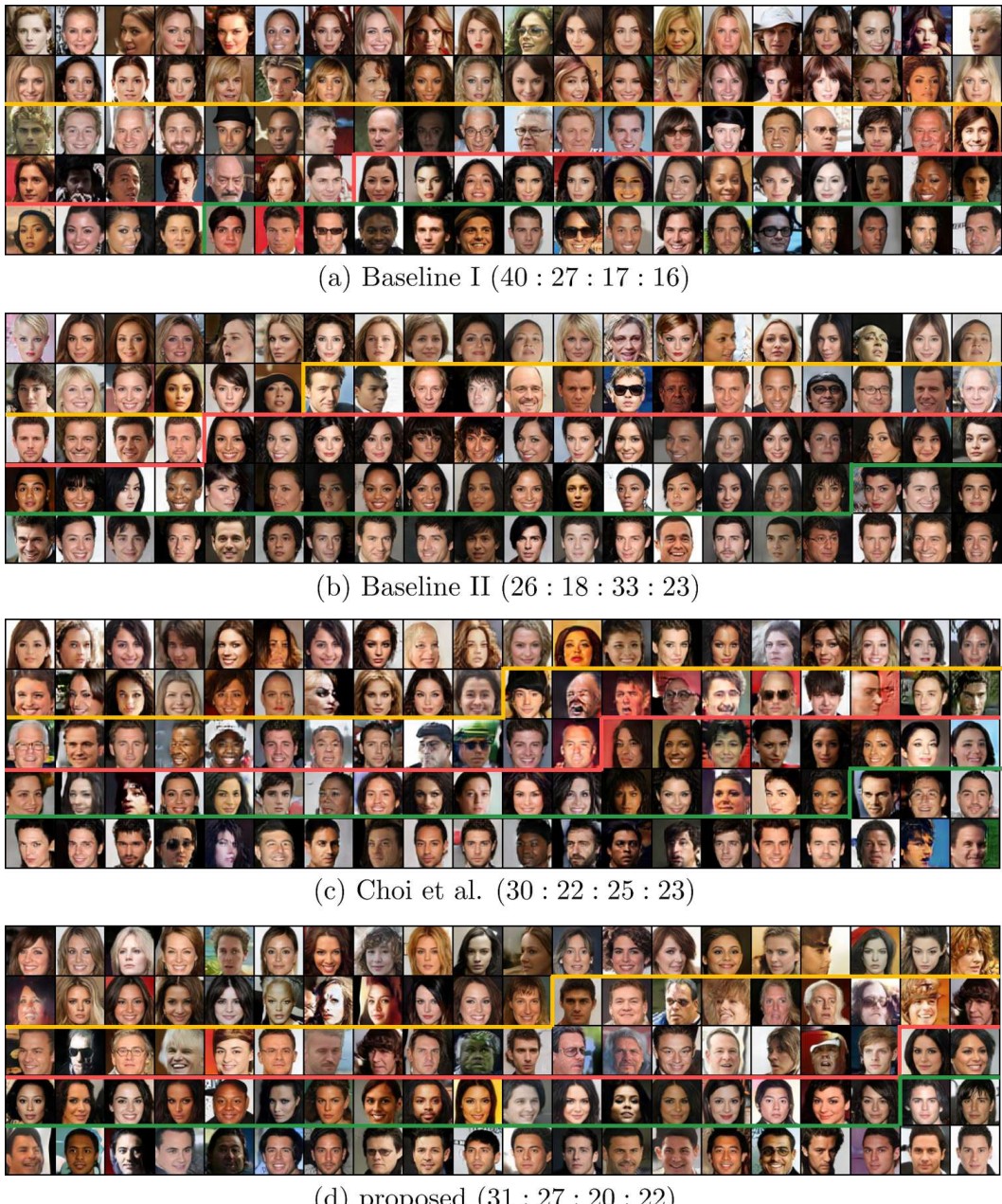

(a) Baseline I $(40 : 27 : 17 : 16)$

(b) Baseline II $(26 : 18 : 33 : 23)$

(c) Choi et al. $(30 : 22 : 25 : 23)$

(d) proposed $(31 : 27 : 20 : 22)$

Figure 7: Generated samples trained on CelebA-multi with 25% reference set size. Faces above the red line are samples without black hair, while the rest are black hair.

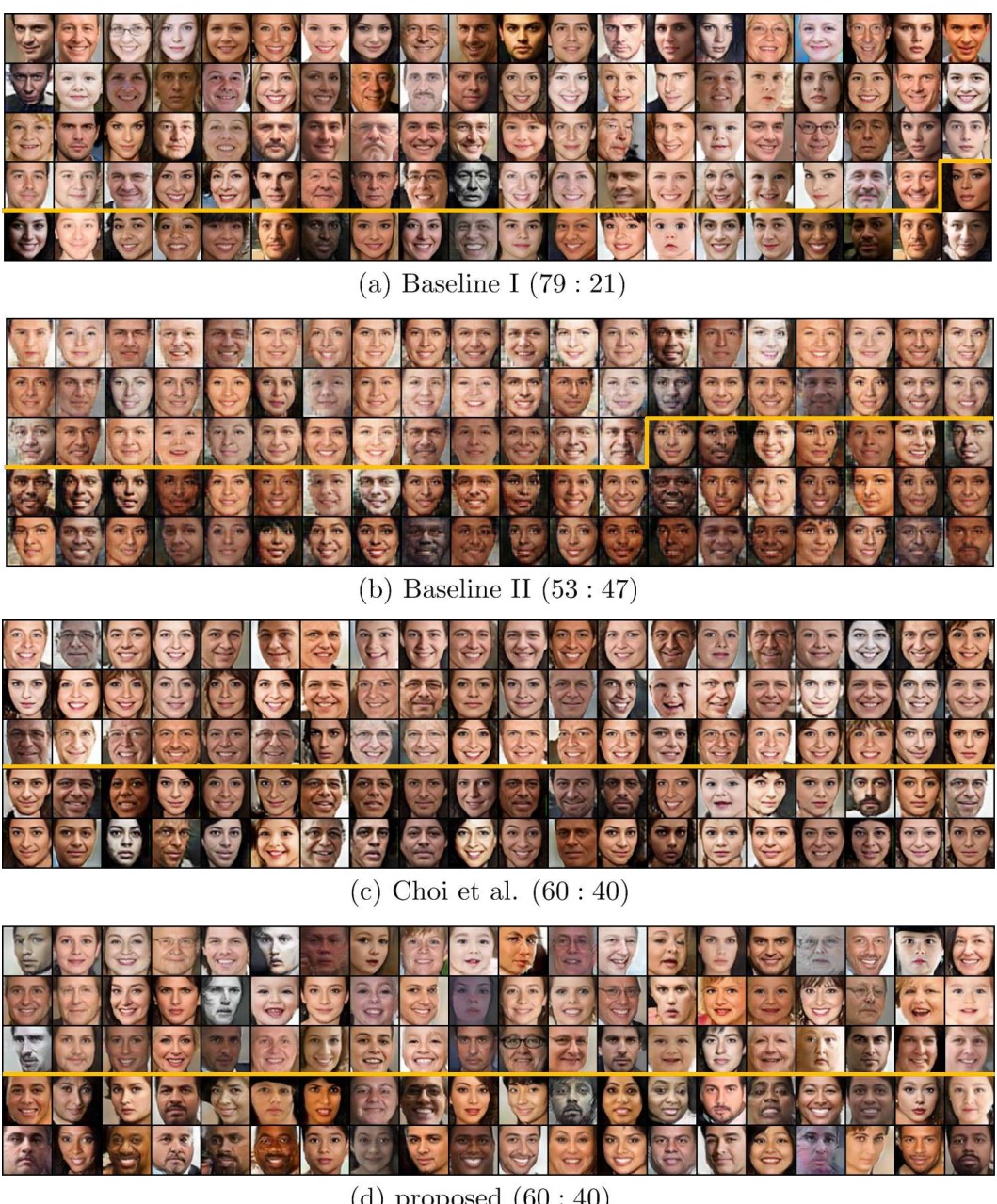

(a) Baseline I (79 : 21)

(b) Baseline II (53 : 47)

(c) Choi et al. (60 : 40)

(d) proposed (60 : 40)

Figure 8: Generated samples trained on UTKFace with 10% reference set size. Faces above the yellow line are white, while the rest are non-white.

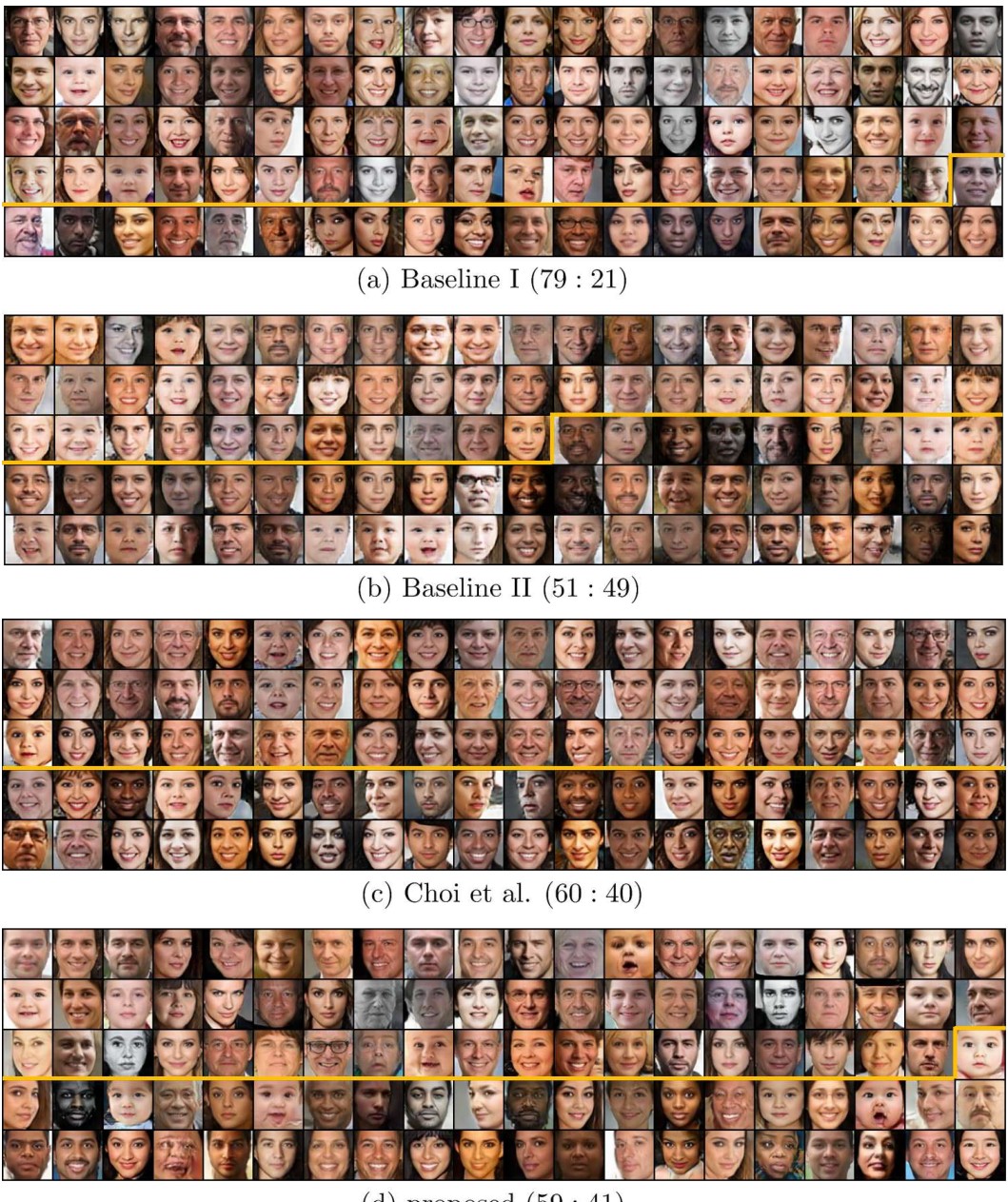

(a) Baseline I (79 : 21)

(b) Baseline II (51 : 49)

(c) Choi et al. (60 : 40)

(d) proposed (59 : 41)

Figure 9: Generated samples trained on UTKFace with 25% reference set size. Faces above the yellow line are white, while the rest are non-white.

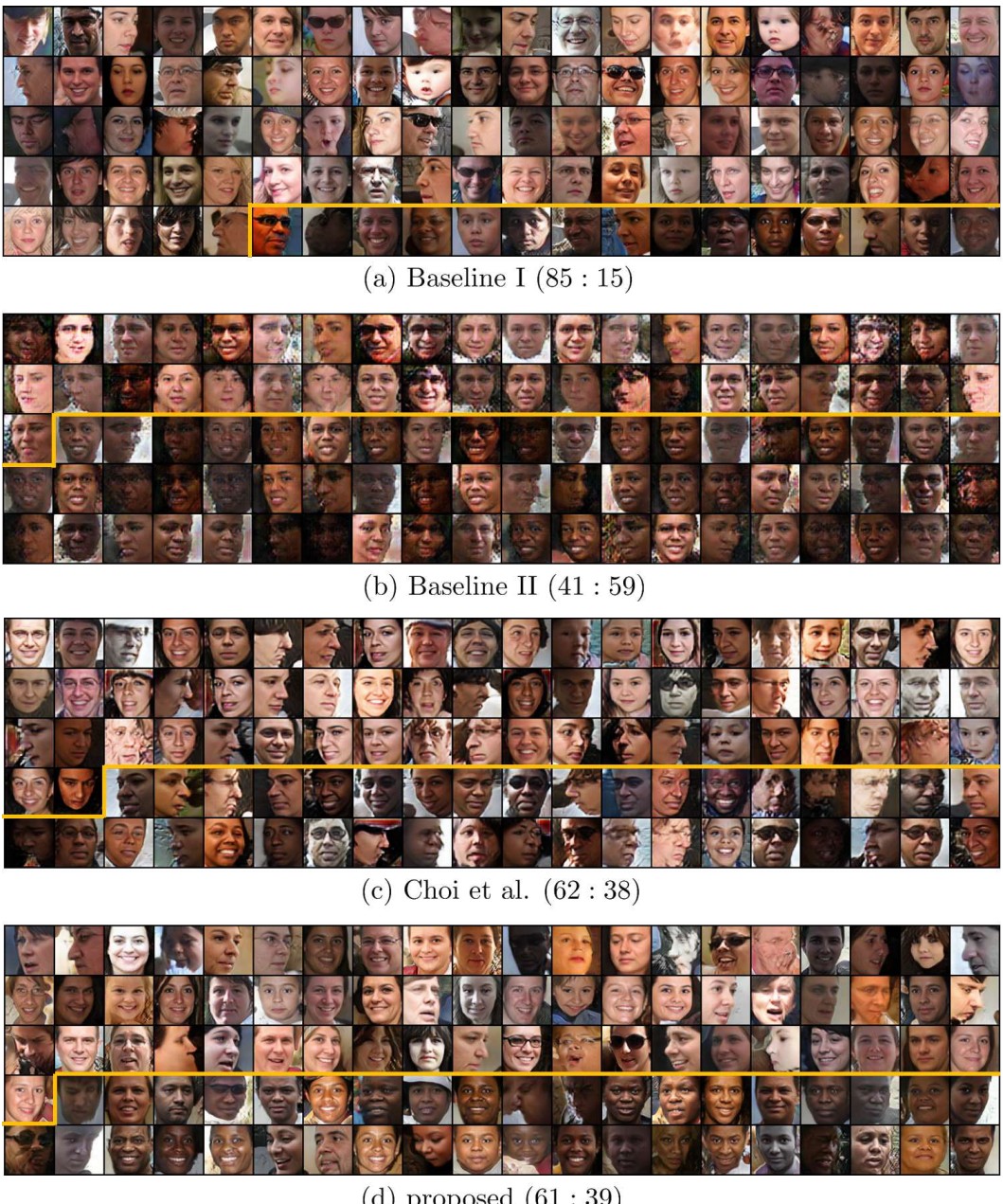

(a) Baseline I (85 : 15)

(b) Baseline II (41 : 59)

(c) Choi et al. (62 : 38)

(d) proposed (61 : 39)

Figure 10: Generated samples trained on FairFace with 10% reference set size. Faces above the yellow line are white, while the rest are black.

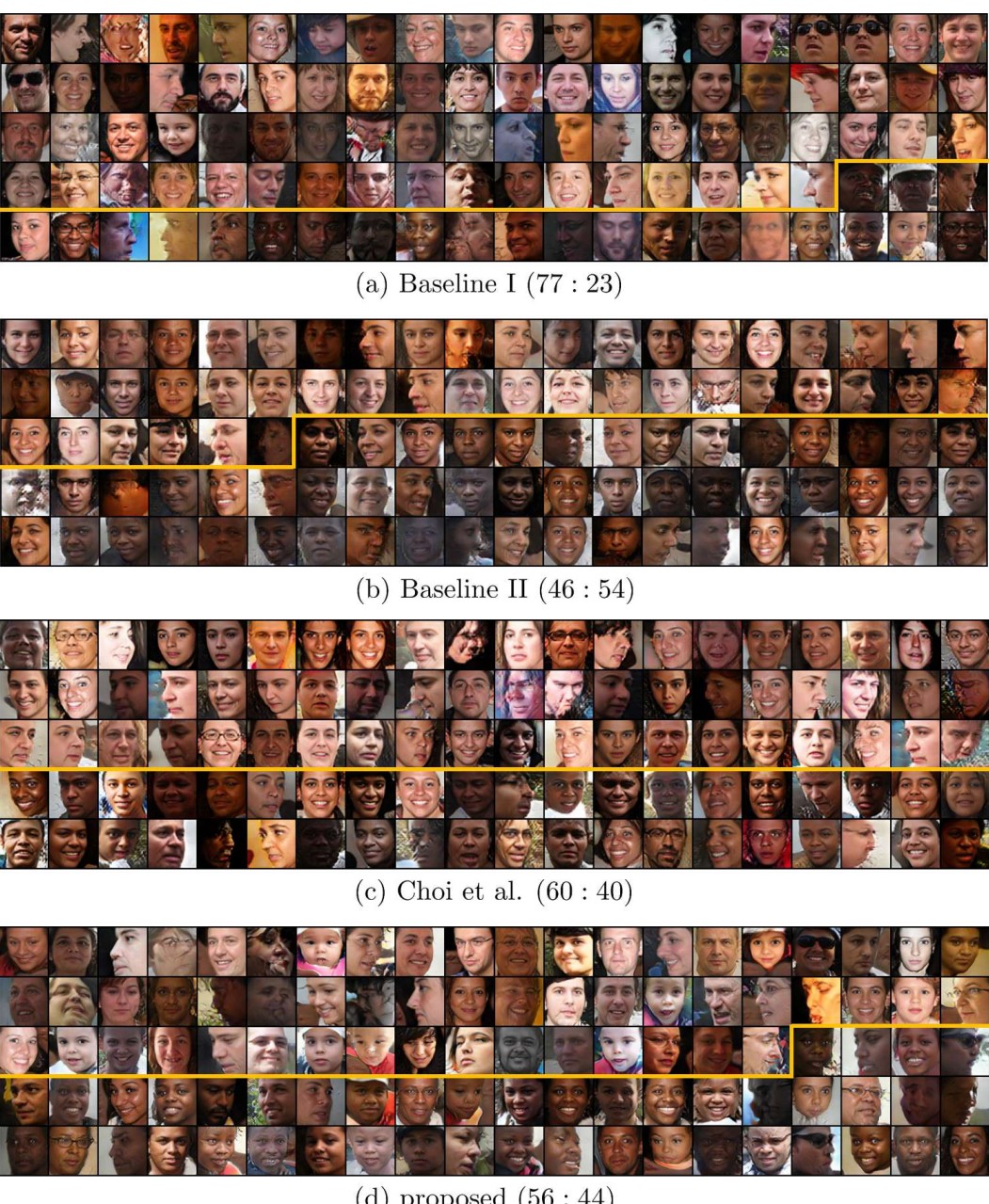

(a) Baseline I (77 : 23)

(b) Baseline II (46 : 54)

(c) Choi et al. (60 : 40)

(d) proposed (56 : 44)

Figure 11: Generated samples trained on FairFace with 25% reference set size. Faces above the yellow line are white, while the rest are black.

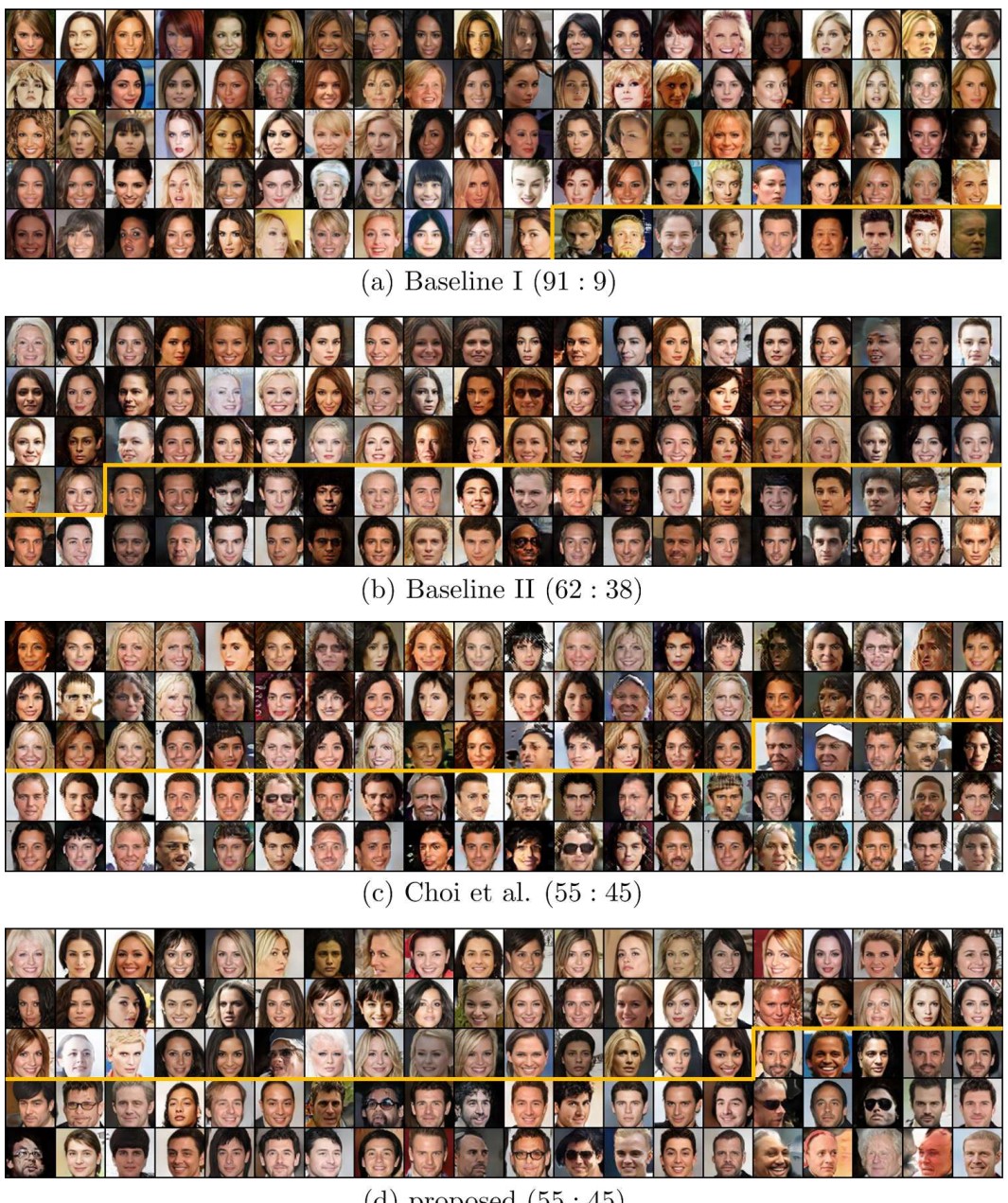

(a) Baseline I (91 : 9)

(b) Baseline II (62 : 38)

(c) Choi et al. (55 : 45)

(d) proposed (55 : 45)

Figure 12: Generated samples trained on CelebA-single with 5% reference set size. Faces above the yellow line are female, while the rest are male.

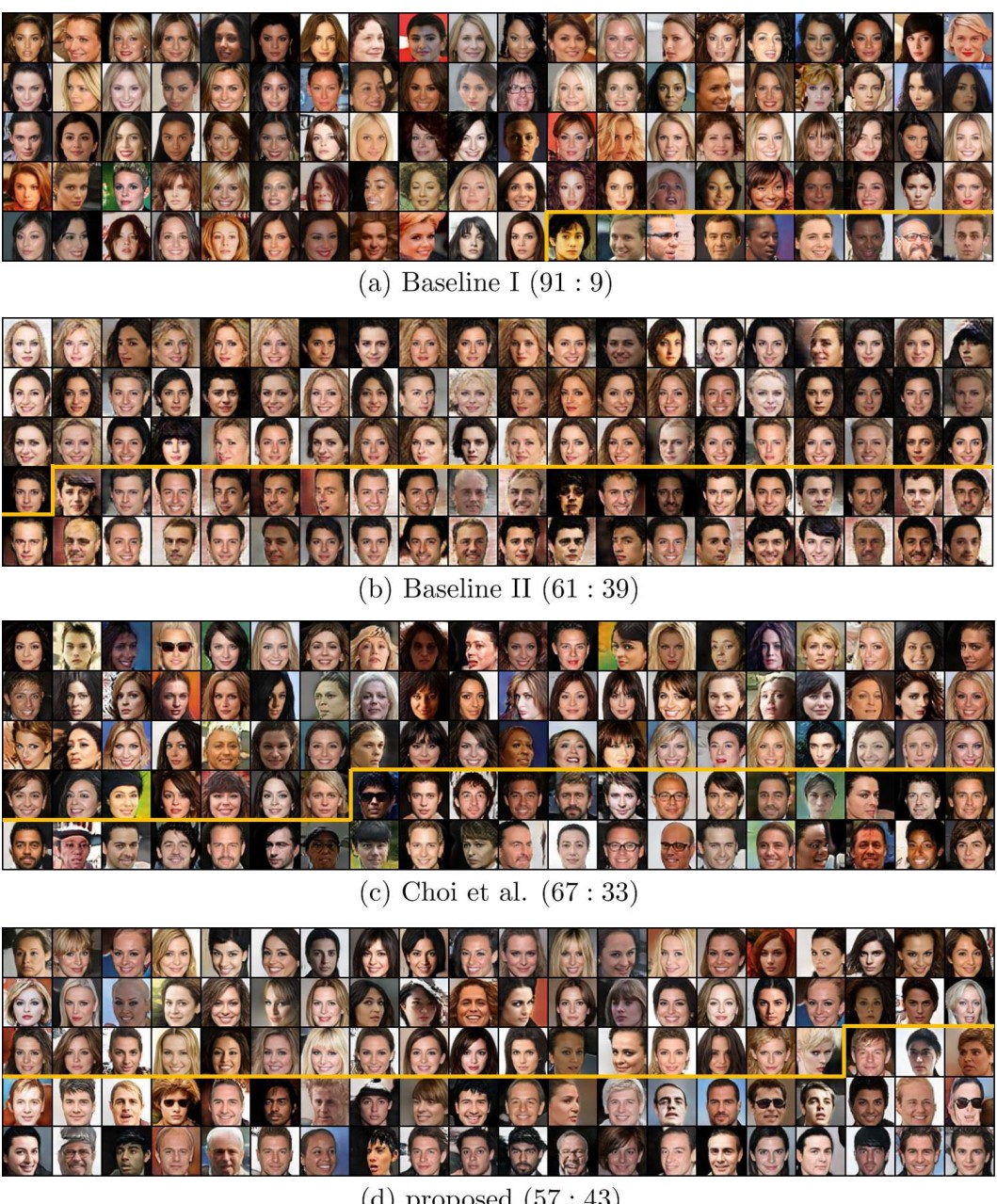

(a) Baseline I (91 : 9)

(b) Baseline II (61 : 39)

(c) Choi et al. (67 : 33)

(d) proposed (57 : 43)

Figure 13: Generated samples trained on CelebA-single with 2.5% reference set size. Faces above the yellow line are female, while the rest are male.

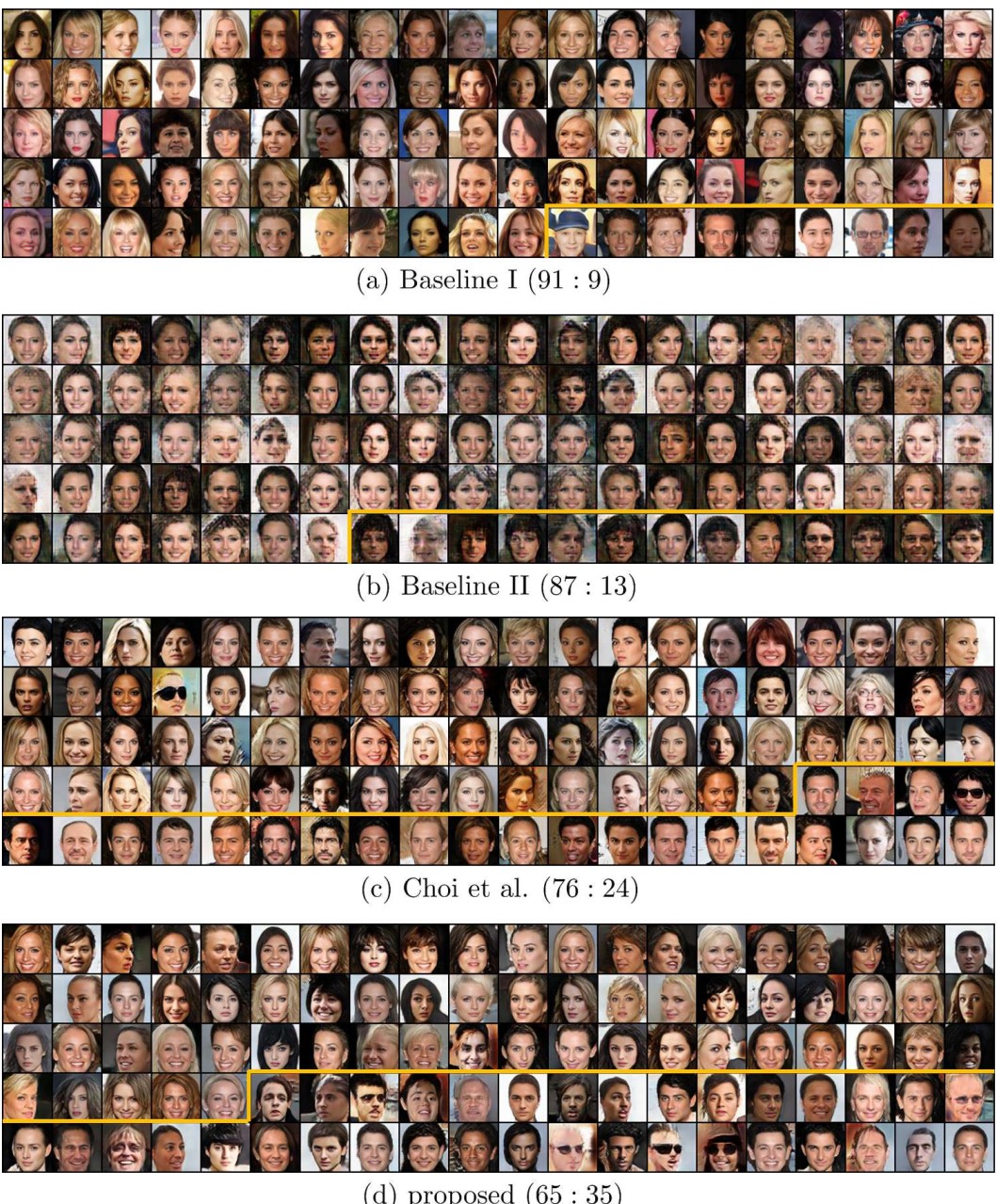

(a) Baseline I (91 : 9)

(b) Baseline II (87 : 13)

(c) Choi et al. (76 : 24)

(d) proposed (65 : 35)

Figure 14: Generated samples trained on CelebA-single with 1% reference set size. Faces above the yellow line are female, while the rest are male.

