# OpenReview forum: "A Fair Generative Model Using Total Variation Distance"
_ICLR.cc/2022/Conference — ICLR 2022 Submitted_

### Official Review · Reviewer_P5wy · 2021-10-28

**Correctness:** 4
**Technical Novelty And Significance:** 2
**Empirical Novelty And Significance:** 2
**Recommendation:** 3
**Confidence:** 3

**Main Review:**

Overall, the paper is well written and easy to follow. The problem of studying fairness in generative models is significant. This paper also conducts an interesting approach by using total variation for regulating “fairness”. However, I have the following major comments.

1. The definition of fairness adopted in this paper is unclear to me and needs further discussion. In particular, I have the following main concerns.

(1) The definition relies on an attribute classifier. What if this classifier itself is biased?

(2) In the same vein, it seems that generative models may produce unrealistic images which are wrongly labeled as female/male by the attribute classifier. How to deal with this situation?

2. My second concern is the comparison with benchmarks in the experiments. In Table 1, the proposed method seems to have a higher discrimination score than Choi et al when the reference size = 25%. Hence, it is unclear if the performance improvement in this paper is significant.

3. The authors observe that “the fairness performance exhibits slight degradation with an increase in the reference set size” in the experiments. However, the explanation given in Appendix C.2 is unclear to me.

4. The authors compare the total-variation-based regularization with other regularization techniques through experiments and demonstrate the results in Table 3. Can the authors provide more implementation details? In particular, which ground metric (i.e., cost function) do they use for the Wasserstein distance? I guess a typical choice is the L2 loss but in the setting of measuring distance between facial images, L2 loss does not seem to be an optimal solution. There are a number of works on ground metric learning for the Wasserstein distance. I wonder what the author did.

#####################
After rebuttal
#####################
I would like to thank the authors for providing a detailed response to my previous comments. However, I still find the fairness definition adopted in this paper a bit strange, especially because it relies on a pre-trained attribute classifier. Hence, I will keep my score unchanged.

**Summary Of The Paper:**

This paper proposes a new generative model with “fairness” constraints by adding a total-variation regularization. The authors conduct several experiments and compare them with benchmark methods.

**Summary Of The Review:**

The definition of fairness adopted in this paper needs further discussion.

---

> ### Author Response · Authors · 2021-11-23
> **Author's response to Reviewer P5wy**
>
> We would like to express our sincere gratitude for your insightful comments. Below we provide details on how we addressed your comments in our revision.
>
> *[4-1] (What if the attribute classifier is biased?):* As you may guess, our fairness definition does not guarantee a faithful evaluation with a biased (or imperfect) attribute classifier. We admit this as a weak point of the fairness notation that we employed herein. In an effort to make the evaluation reliable, we actually employed highly-accurate attribute classifiers in our experiments, around 98% for gender classification for instance. We clarified this point in our revision (2nd paragraph of Section 2, highlighted in blue).
>
> *[4-2] (How to address the situation which yields wrong labeled samples?):* That’s right. Low-quality samples are often wrongly labeled by the attribute classifier. To address this issue, we considered more than 10,000 generated samples for evaluation so as to average out the impact of such wrongly labeled instances. We clarified this point in our revision (2nd paragraph of Section 2, highlighted in blue).
>
> *[4-3] (Marginal improvement over Choi et al.):* Yes, the performance improvement over Choi et al. (2020) is unclear for scenarios where 25% reference set is used. Actually the gain is more noticeable in more practically relevant scenarios having smaller reference set sizes, as exhibited in Tables 1 and 2. Moreover, the proposed method outperforms Choi et al. (2020) in other datasets like UTKFace and FairFace; see Tables 15 and 16 in Section E.1 for details.
>
> *[4-4] (Unclear explanations on why the performance degrades with an increase in the reference set size):* Below we provide a more detailed explanation which are now implemented in the revised Appendix C.2.
>
> For implementing the fairness regularization, we rely upon GAN framework that often yields a model focused on learning only a specific part of representations in a given dataset, rather than covering all the representations inside. Due to this property, with a larger reference set augmented with vast amount of representations, our models (generator and fairness discriminator) can be more encouraged toward learning such augmented (yet less relevant to representations w.r.t. demographic groups) representations and therefore less focused on respecting balanced representations of demographic groups, degrading the fairness performance accordingly. We now included this point in our revision (2nd paragraph of Section C.2, highlighted in blue).
>
> *[4-5] (Implementation details w.r.t. other regularization techniques):* As you expected, we employed the L2 loss for the ground metric of Wasserstein distance. In our revised manuscript, we included this information as well as implementation details w.r.t. other regularization techniques (Section B.2, highlighted in blue).

---

### Official Review · Reviewer_4j8d · 2021-10-29

**Correctness:** 3
**Technical Novelty And Significance:** 3
**Empirical Novelty And Significance:** 3
**Recommendation:** 6
**Confidence:** 3

**Details Of Ethics Concerns:**

I am not an ethics reviewer. This paper provides a method that concerns with Fairness directly. While I do not believe there to be major issues, I believe it would be useful to also have a reviewer (who is more familiar with the ethical aspect of fairness) look at this work.

**Main Review:**

I have enjoyed reading this paper. It is clearly written (though some additional polishing would not hurt), the goal is clearly defined, and the method seems to solve it. However, there still remain some questions from my part which I list below.

## Questions
**[Q1]** Compared to related work, like Choi et al. (2020), the authors do no longer need a classification model (for training). This is a clear step forward from existing work. However, the authors _**do**_ need a "reference distribution/dataset". _Is this not a stricter assumption than needing a classification model?_

I can imagine having a dataset, unlabelled, which I would like to evaluate on its fairness. If I want to quantify exactly how unfair the dataset is, I would need some kind of label to provide a metric of unfairness (which is why the authors still need a classification model to evaluate their model). Using these labels, I can much more easily identify where my data is unfair. For example, if there is a gender discrepancy, a gender-label would indicate this. Using TVD-GAN, we have to assume that the reference dataset does indeed solve for this discrepancy. However, what if besides gender, there is also an ethnic discrepancy? Can I further assume the reference dataset to be fair? Minimising the distributional difference between $\mathbb{P}\_{\text{ref}}$ and $\mathbb{P}\_G$ promotes fairness implicitly, which not only requires additional assumptions about $\mathbb{P}_{\text{ref}}$, but it also seems much less convenient.


**[Q2]** The latter brings me to my second question. _How does this approach relate to domain adaptation and transfer learning?_ The setup detailed in this paper seems very much related. Mainly we wish to learn on one dataset ($\mathcal{D}\_{\text{bias}}$), with a better performance on a second dataset ($\mathcal{D}\_{\text{ref}}$). While there is no explicit prediction target, learning from one "domain" to increase performance in another, seems very much related. Would it be possible to use methods from these fields (e.g. see Table 1 in Wang and Deng (2018) for an overview)?


**[Q3]** There is much work on fairness and generative models. While I don't want to be _that_ reviewer, I do believe some comparison is useful. In particular: FairGAN (Xu et al. (2018)), CFGAN (Xu et al. (2019)), DECAF (van Breugel et al. (2021)), FairGen (Tan et al. (2020)) (which is cited, but not compared against like with Choi et al. (2020)).

## Minor questions
**[MQ1]** In Eq.(5), you introduce a hyperparameter, $\lambda$. Essentially, $\lambda$ controls how fair the eventual generative distribution should be. Higher $\lambda$ indicates to regularise more towards $\mathbb{P}\_{\text{ref}}$, whereas lower $\lambda$ steers the optimisation more towards $\mathbb{P}\_{\text{bias}}$. How does this affect performance (wrt fairness, and FID)?

**[MQ2]** This remark is more a subjective remark, mainly because it concerns Fig.3, which in itself is an almost subjective result. The split between female/male in Fig.3 is quite close between Choi et al. and TVD-GAN. While this in itself is not an issue, it may become one if the generated samples are also quite close to the classification threshold between male and female. Numbering columns as c1, c2, ... and rows as r1, r2, ...; The bottom set includes samples like (r4, c1 -> male), or (r3, c15 -> female), (r4, c5 -> male), (r5, c15 -> male) that could equally well be classified with the opposite gender, making performance the same/worse as Choi et al. I wonder whether tuning $\lambda$, more to $0$ would decrease ambiguity of samples (and how much fairness would suffer as a result)?

**[MQ3]** Perhaps somewhat related to **[Q2]**, but would it not be easier to train only 1 discriminator? First on the biased data, and when converged on the reference data? This would not only make TVD-GAN composable with different notions of fairness (of which there are many), but it would also allow for any trained model to be adjusted when other unfairly distributed attributes need to be dealt with.


_I wish to end my main review by stating that I look forward to discussing further with the authors and reviewers of this paper. My review, nor my score, is set in stone._

### Additional references

Mei Wang, Weihong Deng, _Deep visual domain adaptation: A survey_, Neurocomputing, Volume 312, 2018, Pages 135-153, ISSN 0925-2312, https://doi.org/10.1016/j.neucom.2018.05.083. (https://www.sciencedirect.com/science/article/pii/S0925231218306684)

Depeng Xu, Shuhan Yuan, Lu Zhang, and Xintao Wu. _FairGAN: Fairness-aware generative adversarial networks._ In 2018 IEEE International Conference on Big Data (Big Data), pages 570–575. IEEE, 2018.

Xu, Depeng, Wu, Yongkai, Yuan, Shuhan, Zhang, Lu, & Wu, Xintao. _Achieving Causal Fairness through Generative Adversarial Networks._ Proceedings of the Twenty-Eighth International Joint Conference on Artificial Intelligence, (2019).

Boris van Breugel, Trent Kyono, Jeroen Berrevoets, Mihaela van der Schaar. _DECAF: Generating Fair Synthetic Data Using Causally-Aware Generative Networks_ In Neural Information Processing Systems (2021)






**Summary Of The Paper:**

In their paper, the authors propose a novel generative adversarial network (GAN) which, more than realistic outputs, promotes fairness in the generative distribution. Specifically, when the training dataset is trademarked by some _unfair_ property (the authors' example being unevenly distributed gender or ethnicity), their proposed method-- a "TVD based" approach, which I will call TVD-GAN going forward --will rectify this in their generative distribution. The generative distribution is evaluated using an $L_2$ norm between the likelihoods of some samples given a reference distribution and the generative distribution ($\mathbb{P}_{\text{ref}}$ and $\mathbb{P}_G$, respectively in eq.(1)).

TVD-GAN minimises the "total variational distance" (TVD), through regularisation in eq.(5). As claimed on p.4, TVD "offers the best trade-off performances in fairness and sample quality compared to other divergence measures", as the results indeed seem to suggest. The main goal-- I believe --is to provide a more realistic sample using the larger training dataset, while still maintaining a fair distribution using the smaller reference dataset.

**Summary Of The Review:**

The topic handled in this paper is important, and the contribution is interesting. There are some remaining questions I have regarding the paper in general (listed in my main review above). Given that there are some points the authors have to clarify further, concerning their method as well as related work, I believe a score of 5 is correct at this stage.

However, I do believe the points I raise can be handled during a rebuttal period. Hence, I look forward to discussing more with other reviewers and the authors of this work.

---

> ### Author Response · Authors · 2021-11-23
> **Author's response to Reviewer 4j8d**
>
> We would like to thank your very thoughtful and detailed comments, as well as your useful suggestions, which helped us to improve the manuscript. Below we provide point-by-point responses.
>
> *[3-1] (Classification model vs. reference dataset: which assumption is better?):* We do agree that having a classification model for training (if available) is useful and convenient. However, this assumption is not compatible with our focused problem setting in which demographic labels are not available (which we believe is often the case in reality). On the other hand, as noted in Choi et al. (2020), one can obtain a reference dataset without demographic labels by taking some carefully-designed data collection protocols from organizations like World Bank and biotech firms [1-2]. We clarified all of these points in our revision; see the 5th and 1st paragraphs of Sections 1 and 2, respectively (highlighted in blue).
>
> *[3-2] (Relation to domain adaptation and transfer learning):* We do agree that our problem setting can be interpreted as a transfer learning framework. However, we believe our approach is inherently distinct from the methods in the transfer learning literature, and employing such methods (including the ideas in the referred paper (Wang and Deng (2018)) in our context seems not that straightforward. We included an in-depth discussion on this point in our revision (Section C.3, highlighted in blue).
>
> *[3-3] (Comparison with existing fair generative models):* As noted in the related works in Section 1, the setting considered in our study is distinct from those in the referred papers. They either: (1) focus on a different fairness notion (FairGAN (Xu et al. (2018)), CFGAN (Xu et al. (2019)), and DECAF (van Breugel et al. (2021)); or (2) rely upon another assumption which is not compatible with our setting (FairGen (Tan et al. (2020)). Hence, we believe direct comparisons cannot be made to those works. We do think the referred papers are indeed relevant; hence now cited them in our revision (5th paragraph of Section 1, highlighted in blue).
>
> *[3-4] (Impact of $\lambda$ on fairness and intra FID):* We believe Fig. 4 in Section E.4 is exactly what you wanted to see, where we provide a trade-off curve that exhibits the impact of adjusting $\lambda$ on our twin performance metrics: fairness discrepancy and intra FID. Notice in Fig. 4 that as $\lambda$ increases, fairness performance improves (smaller fairness discrepancy) at the expense of the degraded sample quality, reflected in larger intra FID for both demographic groups.
>
> *[3-5] (Re. Figure 3):* We do agree that Fig. 3 does not clearly illustrate whether our approach outperforms Choi et al. (2020) in fairness. Instead, Table 1 provides a more precise comparison for the same scenario (CelebA-single with 10% reference set), where the fairness performances are evaluated with more than 10,000 generated samples. Notice in Table 1 that the proposed approach exhibits better fairness performance than Choi et al. (2020) on CelebA-single with 10% reference set, reflected in lower fairness discrepancy. We clarified this point in our revision (2nd paragraph in Section 4.2, highlighted in blue). Re. whether tuning $\lambda$ decreases the ambiguity of samples: Yes, we observed that decreasing $\lambda$ indeed decreases the ambiguity in our experiments. Refer to Fig. 4 in Section E.4 for the tradeoff relationship between fairness and sample quality.
>
> *[3-6] (Suggestion of using only one discriminator):* As per your great suggestion, we now conducted new experiments to evaluate performance for the case having only one discriminator. Below we compare the performance with our approach, evaluated on CelebA-single with 10% reference set:
>
> |Method|Male_FID|Female_FID|Fairness|
> |------|---|---|---|
> |One disc.|19.79 ± 0.451|13.40 ± 0.514|0.147 ± 0.008|
> |Proposed|14.29 ± 1.354|9.51 ± 1.069|0.057 ± 0.012|
>
> Notice that the proposed approach outperforms the one discriminator method both in fairness and sample quality. We now included this result in our revision (Section C.3, highlighted in blue).
>
> ### References
> [1] The real issue: Diversity in genetics research, 23andMe Blog, 2016.
>
> [2] 23andme has a problem when it comes to ancestry reports for people of color, Quartz, 2016.

---

> > ### Comment · Reviewer_4j8d · 2021-11-25
> > **Dear Authors**
> >
> > Dear authors,
> >
> > Thank you for responding to my review. I mostly agree with the responses you provided to my questions. There a some remaining.
> >
> > _[3-3] (Comparison with existing fair generative models)_ Could you explain how their fairness notion(s) are different than the one you use in your manuscript? As I understand, some of these methods are used in a tabular setting, which could limit their use in your setting; though the notion of "fair" data seems (should be?) transferable across domains.
> >
> > _[3-5] (Re. Figure 3)_ While I don't consider this a _dealbreaker_, I would be interested to somehow see the presented results relate to the classifier performance. Since the proposed method performance closely resembles that of Choi et al.; a lot seems to depend on the border cases, which any classifier would struggle with. Would it be possible to see a p(Gender | X) type output per image in Fig.3? I realise we passed the deadline for you to update the manuscript, so a discussion on this would suffice.
> >
> > _[3-6]_ Thanks for this.

---

> > > ### Author Response · Authors · 2021-11-29
> > > **To Reviewer 4j8d**
> > >
> > > Thanks for asking insightful questions. Please see below for the responses:
> > >
> > > *[3-3] (Comparison with existing fair generative models):* The fairness notion considered in the referred papers (FairGAN (Xu et al. (2018)), CFGAN (Xu et al. (2019)), and DECAF (van Breugel et al. (2021)) is about independence between decision and demographic labels, and the goal therein is to guarantee that the generated decision labels are (statistically) independent of the given demographic labels. Note that this notion is not only distinct from the fairness notion that we focused on herein, but also incompatible with our problem setting in which demographic labels (as well as decision labels) are not available, and therefore we believe the transferring is not possible.
> > >
> > > *[3-5] (Re. Figure 3):* We cannot provide the requested quantities, as we did not save the raw data of images in Fig. 3, and those are randomly-generated samples which are almost impossible to reproduce all of them. Instead, we can share the values of p(Gender | x) with 10,000 newly generated images, which enables a more precise assessment on the impact of the border-case instances that you are concerned about. Below we provide the fractions of samples w.r.t. the range of p(Gender | x) of the two approaches, regarding 10,000 generated samples for each method:
> > >
> > > | $\mathbb{P}({\hat Z}=1 \| x)$ | $[0.0, 0.2)$ | $[0.2, 0.4)$ | $[0.4, 0.6)$ | $[0.6, 0.8)$ | $[0.8, 1.0]$ |
> > > |------|:---:|:---:|:---:|:---:|:---:|
> > > | Choi et al. | 0.5282 |  0.0312 | 0.0238 | 0.0257 | 0.3911 |
> > > | Proposed | 0.5113 | 0.0214 | 0.0159 | 0.0179 | 0.4335 |
> > >
> > > Notice that the impact of such border-case instances (whose p(Gender | x) is near 0.5) is not that significant in both approaches, for around 2% of the total number of samples.

---

> > > > ### Comment · Reviewer_4j8d · 2021-11-29
> > > > **Dear Authors**
> > > >
> > > > Dear authors,
> > > >
> > > > Thank you for addressing my remaining points. I have updated my score to an accepting score.
> > > >
> > > > 4j8d

---

### Official Review · Reviewer_VNCK · 2021-10-31

**Correctness:** 2
**Technical Novelty And Significance:** 2
**Empirical Novelty And Significance:** 2
**Recommendation:** 3
**Confidence:** 5

**Main Review:**

Even though the paper studies an interesting topic and tries to address a vital problem, in its current form the paper is not ready to go under revision. The paper is poorly written. The authors may find below my major and minor comments.

## Major Comments:

1) The optimization problem in~(6) is not a three-player game. Two players that are maximizing are collaborating. Therefore, this is still a 2-player game. It would have been a 3-player game if the optimization problem was in the form of min-max-min, which is not the case.

2) Why does the TVD regularization offer the best trade-off performances in fairness and sample quality compared to other divergence measures?

3) Algorithm 1 proposes to use stochastic gradient descent with mini-batch. Are the gradients of the objective are unbiased?

4) I am not sure how does reference dataset is collected in real-life? How one can ensure that the reference dataset is balanced in real-life?

5) I do not fully understand why increasing the size of the reference set has a detrimental effect on the performance of the framework?  Can you provide more additional insights into the explanations in Appendix C.2?

## Minor Comments:

1) I am not sure what does the first sentence of the contributions section means. Which key benefit feature is mentioned in~Roh et al.?
2) The paper is not self-contained. What is the attribute classifier introduced in Choi et al. (2020)?
3) The first paragraph of Section 3.1 is very badly written. I would suggest the authors improve the English of this sentence.

**Summary Of The Paper:**

This paper studies fairness in generative models. When the training data is biased, it may result in imbalanced generated samples. They assume that they have access to scarce reference samples that are balanced.
They measure the unfairness of the model through total variation and propose a fairness-aware model that is regulated by total variation distance between the generated samples (biased) and reference samples (unbiased). The authors further assume that the sensitive attributes are not available. Finally, they provide numerical experiments to demonstrate the performance of their proposed framework.

**Summary Of The Review:**

This paper is not ready to go under review. I think this paper can be further improved in terms of writing. Once, it is written at an academic level it can go under review. I would suggest the authors improve the presentation of their paper.

---

> ### Author Response · Authors · 2021-11-23
> **Author's response to Reviewer VNCK**
>
> We would like to express our sincere gratitude for providing insightful comments and useful suggestions, which indeed helped us to improve the manuscript. Below we provide point-by-point responses.
>
> *[2-1] (Re. equation 6):* As noted in Remark 2, the two discriminators in (6) interact indirectly through the generator G, and we observe that the interaction is in a negative manner. This is because encouraging balanced samples (the goal of $D_{\sf fair}$) may yield less realistic samples (the opposite goal of $D$) due to the small-sized reference set. Hence, we believe the optimization in (6) is indeed a three-player game.
>
> *[2-2] (Why TVD performs the best compared to other divergence measures):* Actually, we do not have a clear answer. Instead we can share some yet subtle insights from one recent work [1] which demonstrated that TVD offers more robust performance to the size of dataset, relative to other measures like KL and Chi-square; see Section 3.3 in [1] for details. We clarified this point in our revision (1st paragraph of Section 3.2, highlighted in blue).
>
> *[2-3] (Re. the stochastic gradients in Algorithm 1):* Yes, the stochastic gradients appeared in Algorithm 1 are indeed unbiased w.r.t. the true gradients of the objectives.
>
> *[2-4] (How to collect balanced reference dataset in real-life):* As in Choi et al. (2020) which firstly introduced balanced reference data, we hereby provide a concrete example of such reference data. Organizations like World Bank and biotech companies [2,3] often implement a set of good practices on their collected data to come up with a representative dataset, wherein one can expect balanced representations over a variety of demographic groups. We believe such representative dataset is one instance of a reference set. We now included this example in our revision (1st paragraph of Section 2, highlighted in blue).
>
> *[2-5] (Unclear explanations on why increasing the reference set size has a detrimental effect):* For implementing the fairness regularization, we rely upon GAN framework that often yields a model focused on learning only a specific part of representations in a given dataset, rather than covering all the representations inside. Due to this property, with a larger reference set augmented with vast amount of representations, our models (generator and fairness discriminator) can be more encouraged toward learning such augmented (yet less relevant to representations w.r.t. demographic groups) representations and therefore less focused on respecting balanced representations of demographic groups, degrading the fairness performance accordingly. We now included this point in our revision (2nd paragraph of Section C.2, highlighted in blue).
>
> *[2-6] (Unclear sentence in the contribution paragraph):* What we meant by the key benefit of Roh et al. (2020) is that their robust training method (employing divergence-based regularization) enables a more efficient use of clean reference data, as compared to reweighting-based robustness approaches. We clarified this point in our revision (3rd paragraph in Section 1, highlighted in blue).
>
> *[2-7] (What is the attribute classifier introduced in Choi et al. (2020)?):* It simply denotes a pre-trained classifier for predicting a sensitive attribute. We clarified this point in our revision (2nd paragraph of Section 2, highlighted in blue).
>
> *[2-8] (A poorly written paragraph in Section 3.1):* As per your suggestion, we tried to polish the writing of the part in our revision; see the changes highlighted in blue.
>
>
> ### References
> [1] Regularizing Generative Adversarial Networks under Limited Data, CVPR 2021.
>
> [2] The real issue: Diversity in genetics research, 23andMe Blog, 2016.
>
> [3] 23andme has a problem when it comes to ancestry reports for people of color, Quartz, 2016.

---

### Official Review · Reviewer_E8Px · 2021-11-03

**Correctness:** 3
**Technical Novelty And Significance:** 2
**Empirical Novelty And Significance:** 2
**Recommendation:** 5
**Confidence:** 4

**Main Review:**

Overall, the idea and approach are clearly presented. The experiment section also contains multiple different empirical studies. It would be very helpful if the authors can kindly clarify the following points:

### q1: the utilization behind the distance metric w.r.t. the practical scenario described

In the abstract, if I understand it correctly, the practical problem that motivated the study is the unavailability of demographic group labels. It is not entirely clear to me how the technique (of fair generative model w/ the TV distance metric) can help solve this practical issue. Please kindly let me know if there is any misunderstanding.

### q2: regarding Theorem 1

While I understand the content of the theorem itself, I am wondering if the denotation of "(Nowozin et al., 2016)" is trying to indicate the fact that Theorem 1 is already proposed in the referred paper. Please kindly clarify this as this theoretical result seems to be an essential part of the contribution (apart from the empirical findings).

### q3: regarding experiments

- Table 2, Baseline II, 2.5% reference set size: I am wondering if the authors could kindly share some insights regarding why the fairness metric is so small (seems to be even better than the proposed approach in all listed cases).
- I am wondering if the TV measure has anything to do with the sample size of the reference set. With a very big size difference (e.g., 1% reference set size), how can we justify that TV distances are well-estimated?
- Figure 3 shows the generated images, both in terms of the representation balance and the quality. I am wondering if the authors can share some insights regarding why the proposed approach can (1) generate better-quality images (2) use weaker supervision (smaller reference sets) at the same time. Is it all about the TV metric, which does not seem to be the case based on the msg conveyed by Table 3, or something else?

**Summary Of The Paper:**

This paper focuses on the fair generative model problem. In particular, "fairness" lies in the balance of representation among groups in the data. The paper proposes to use TV as the distance metric in order to capture the difference between the generated distribution and the reference distribution. Empirical results are also provided.

**Summary Of The Review:**

The paper proposes to utilize TV distance in the task of fair representation learning. The empirical experiments suggest a strong performance. It would be very helpful if the authors can kindly comment on the questions in the `Main Review`, so that the significance of the results can be better appreciated.

---

> ### Author Response · Authors · 2021-11-23
> **Author's response to Reviewer E8Px**
>
> *[1-1] (The utilization behind the distance metric w.r.t. the practical scenario described):* Yes, the motivation of our study came from the unavailability of demographic labels which often holds in practice. However, our TVD-based approach does not attempt to address the practical issue. It rather provides a way to regulate a fairness issue under such realistic assumption with an access to small balanced reference set. We clarified this point in our revision (the abstract, highlighted in blue).
>
> *[1-2] (Re. Theorem 1):* As you may guess, Theorem 1 is a known result presented in Nowozin et al. (2016) and therefore not our contribution. We further clarified this point in our revision (1st paragraph of Section 3.1 (highlighted in blue).
>
> *[1-3] (Why the fairness metric is so small for Baseline II with 2.5% reference set):* We do appreciate sharing your sharp observation. We found that the value you pointed out is misreported; please see the following for the correct one: 0.150 ± 0.003. We corrected this error in our revision (Table 2 in Section 4.2, highlighted in blue).
>
> *[1-4] (Connection between the goodness of TVD measure and the size of reference set):* Yes, there is a connection indeed. If a given reference dataset is too small to well-represent the underlying reference distribution, the measured TVD (via such small-sized reference set) would then be inaccurate, degrading the fairness performance accordingly. We clarified this point in our revision (1st paragraph of Section C.2, highlighted in blue).
>
> *[1-5] (Why the proposed approach is more robust to the reference set size than Choi et al. (2020)):* We believe the benefit of our approach comes from the use of two key ingredients: (i) divergence-based regularization; (ii) TVD measure. As mentioned in Section 1, a divergence-based regularization technique can yield an efficient use of small reference data. On the other hand, as exhibited in [1], TVD measure is robust to the size of considered dataset compared to a set of f-divergences like KL and Chi-square. We clarified this point in our revision (1st paragraph of Section 3.2, highlighted in blue).
>
> ### Reference
> [1] Regularizing Generative Adversarial Networks under Limited Data, CVPR 2021.

---

### Public Comment · ~Choi_C1 · 2022-06-20
**Discrepancy in Experimental Results when Compared with Original Paper (Choi's et.al work)**

Dear Authors,

Thank you for the interesting contribution in the field of fair generative models.  We would like to clarify some implementation details for the experiments seen in table 1 and table 2.

We notice that there are discrepancies in the baseline measurements for your experiments, where your replicated experiments demonstrate considerable improvement in Choi’s et.al work ([https://arxiv.org/pdf/1910.12008.pdf](https://arxiv.org/pdf/1910.12008.pdf)). We had attempted to repeat the experiments with the source code you had provided at perc={0.1,0,25}, but were unable to replicated these results. We followed the provided instructions religiously and even utilised the provided hyper parameters script. The results were most troubling at perc=0.1, where the GAN did not converge. Our results are in the the following tables.

$\underline{Perc=0.25\  and\ bias=(0.9,0.1)}$
|           | Original Result Choi et.al | Reported Results in this work | Replicated results by the provided code |   |
|-----------|----------------------------|-------------------------------|-----------------------------------------|---|
| Intra FID |  ---                           | 20.68                         |     ---                                      |   |
| FID       | $\approx 28$               |  ---                               | 24.25                                   |   |
| FD        | $\approx 0.45$             | 0.065                         | 0.15                                    |   |

$\underline{Perc=0.1\ and\ bias=(0.9,0.1)}$
|           | Original Result Choi et.al | Reported Results in this work | Replicated results by the provided code |   |
|-----------|----------------------------|-------------------------------|-----------------------------------------|---|
| Intra FID | ---                             | 25.74                         | ---                                          |   |
| FID       | $\approx 25$               |---                                 | 304                                     |   |
| FD        | $\approx 0.1$              | 0.104                         | 0.5                                     |   |

Additionally we notice that there are some modification in the provided code e.g., addition of CNN3/CNN5 and temperature calibration in the density ratio classifier, we were wondering if more insight could be provided as to how these results were produced and if any additional modifications were made?

---

> ### Public Comment · ~Soobin_Um1 · 2022-06-23
> **Response to Choi: Clarifying some implementation details**
>
> Dear Choi,
>
> Thanks for your attention to our work.
>
> While reproducing the method proposed in Choi et al. (2020), we found that employing simpler architecture (like CNN3/CNN5 that you mentioned) for density ratio classifier greatly improves performance compared to the results reported in Choi et al. (2020), especially when given reference datasets are small (e.g., 10% ref. set). Also, for training BigGAN, we found the use of both larger alpha values and smaller batch sizes leads to better performances. In particular, our choices (on the above settings) for your concerned scenarios were:
>
> (i) perc 0.1:
> - DR clf: CNN3 with ncf=64
> - BigGAN: batch_size = 8, alpha = 1.0
>
> (i) perc 0.25:
> - DR clf: ResNet18
> - BigGAN: batch_size = 8, alpha = 1.5
>
> For the other parameters, we took the same settings provided in Choi et al. (2020).
>
> Hope this helps your concerns.

---

### Decision · Program_Chairs · 2022-01-20

**Decision:**

Reject

**Comment:**

This paper is concerned with fairness in the generative setting, specifically the setting in which various groups have very different sizes, and are therefore treated disproportionately by the model, with the group memberships further being unknown.

The reviewers generally agreed that the setting was interesting and important. However, they were critical of the writing quality, significance, and quality of the theoretical contribution.

The authors made significant improvements in the review period, and while these were not quite enough to satisfy enough reviewers, opinions clearly changed in a positive direction during the discussion period. Future changes motivated by the existing reviewer concerns should significantly improve the paper.